# Graphene: A Path-Breaking Discovery for Energy Storage and Sustainability

**DOI:** 10.3390/ma15186241

**Published:** 2022-09-08

**Authors:** Deepam Goyal, Rajeev Kumar Dang, Tarun Goyal, Kuldeep K. Saxena, Kahtan A. Mohammed, Saurav Dixit

**Affiliations:** 1Chitkara University Institute of Engineering and Technology, Chitkara University, Rajpura 140401, India; 2Department of Mechanical Engineering, University Institute of Engineering and Technology, Panjab University SSG Regional Centre, Hoshiarpur 146021, India; 3Department of Mechanical Engineering, IK Gujral Punjab Technical University, Jalandhar 144603, India; 4Department of Mechanical Engineering, GLA University, Mathura 281406, India; 5Department of Medical Physics, Hilla University College, Babylon 51002, Iraq; 6Peter the Great St. Petersburg Polytechnic University, 195251 St. Petersburg, Russia; 7Division of Research & Innovation, Uttaranchal University, Dehradun 248007, India

**Keywords:** graphene, renewable energy, solar cells, batteries, fuel cells, nanolubricants, supercapacitors, sustainability

## Abstract

The global energy situation requires the efficient use of resources and the development of new materials and processes for meeting current energy demand. Traditional materials have been explored to large extent for use in energy saving and storage devices. Graphene, being a path-breaking discovery of the present era, has become one of the most-researched materials due to its fascinating properties, such as high tensile strength, half-integer quantum Hall effect and excellent electrical/thermal conductivity. This paper presents an in-depth review on the exploration of deploying diverse derivatives and morphologies of graphene in various energy-saving and environmentally friendly applications. Use of graphene in lubricants has resulted in improvements to anti-wear characteristics and reduced frictional losses. This comprehensive survey facilitates the researchers in selecting the appropriate graphene derivative(s) and their compatibility with various materials to fabricate high-performance composites for usage in solar cells, fuel cells, supercapacitor applications, rechargeable batteries and automotive sectors.

## 1. Introduction

Industrialization, globalization, urbanization and population explosion have put extensive pressure on global energy, and in order to meet demand and supply the required energy, there is a need to explore sustainable and renewable forms of energy [1,2]. Focus has to be placed on the development of materials, mechanisms, methods and, above all, a positive mind-set of human beings toward energy conservation. Researchers across the globe have come out with innovative materials which are capable of transforming the vast energy sector, and one such material, graphene (GR), has been at the forefront of industry and academic research since 2010, when the Nobel Prize was bestowed upon physicists for ground-breaking experiments on this super material. There has been enormous work on patents, research publications, industry projects and applications in different domains with graphene as the center theme. Although, extensive research is being conducted on the multifarious uses of GR, energy conservation and storage are the most critical for the global sustainable economy. There are a variety of materials which can be put to use for energy storage, but choice is limited when cost and energy-to-weight ratio are taken into consideration [3,4]. Carbon, with its property of high surface area, is the lightest material which can be used for energy storage [5]. Being a non-metal, carbon (C) is an environmentally friendly and inexpensive material having an atomic number 6 configuration of 1s^2^2s^2^2p^2^. Carbon is available in diverse allotropic forms, and graphite is considered as one of the most important forms. It is in the form of a three-dimensional allotrope, in which layers are parallel stacked with carbon atoms, hybridizing sp^2^. Strong bonding exists between carbon atoms in an individual layer, but layers are held to one another by weak Van der Waals force [6]. One individual layer is termed as graphene. An analogy can be made by considering graphite as a complete book and GR layers as pages of that book. No conventional material possesses such properties or can replace graphene in multifarious applications [7]. Since the inception and synthesis of GR by the ‘Scotch tape method’, new doors in this active field of research have opened [8,9]. This lightweight, flexible and resistant material was born out of the GR Flagship research program in Europe, and it was possible to see GR with the naked eye because of its ability of absorbing 2.3% light [10]. Table 1 summarizes the properties of the thinnest yet strongest material—‘graphene’.

GR, with a harder structure than diamond and nearly 300 times stronger than steel, is also an excellent heat conductor which surpasses diamond. GR is capable of being coated on different materials and can be stretched with ease. Such extraordinary properties can be utilized in the diverse fields as energy storage, the electrical and electronics sectors, solar cells, the aerospace and automotive sectors, telecommunication, and the medical field [14]. Figure 1 summarizes the interrelationship between GR properties and their usage in different energy devices. Moreover, in solar air heaters, graphene has also been widely used as a coating material on absorber plates to increase their performance [15,16]. Graphene oxide (GRO)-based adsorbents functions effectively in removing water pollutants [17,18].

## 2. Graphene Synthesis

There has been ample interest in the development of nanotubes, fullerenes, and graphite oxide/graphitic oxide, also termed “graphitic acid”, since long ago. Extensive efforts and research have been carried out to produce GR, which should be of excellent quality and have the capability to be used in different applications, on a large scale [19]. In 1859, graphite was treated with a solution consisting of fuming nitric acid and potassium chlorate by the chemist Brodie from Oxford in order to prepare graphitic acid [20]. About four decades after this invention, Staudenmaier [21] carried out an improvement in the oxidation technique by adding KClO_3_ in restricted quantity along with concentrated sulphuric acid instead of addition in a single step. This improved technique led to achieving the same ratio of carbon and oxygen (2:1) as in Bordie’s experiments, but with the advantage of avoiding repeated multiple oxidations. It is evident from the literature that P.R. Wallace had predicted the presence of electronic properties in single-layered graphite (nowadays termed as GR) in the year 1940, and this has now been confirmed. In 1975, Lang et al. [22] attempted the synthesis of monolayer graphite and demonstrated the building of multi- as well as mono-layered graphite using platinum substrates and thermally decomposing carbon over these. There was inconsistency in the properties of these sheets, and so the process was not studied in detail and the benefits of this product were not explored at that time. In 1999, scattered attempts were made to produce GR [23,24], and, in 2004, Novoselov et al. [7,25] was credited for the invention of GR. They showed repeatable synthesis of GR for the first time through exfoliation, and this material came under the class of specialized nanomaterials. GR, with a zero-band gap, has normally been grouped in the category of semi-metals with nano/microstructure, and is of extensive use in industry [26,27].

Small samples of GR can be made by a simple and economical method, termed the ‘adhesive tape method’, during which micromechanical peeling of highly ordered pyrolytic graphite (HOPG) occurs [8,28,29]. These processes, quite analogous to skin exfoliators, are useful for obtaining pristine graphene flakes, which can be useful for research purpose. However, such a technique is time consuming and inappropriate for large-scale production, and the flake size achieved in the micrometer range [8,30]. Work has been done on bulk production of GR by epitaxially growing thin graphitic films on silicon carbide in the temperature range of 1250–1450 °C [31,32]. Although this approach results in high quality graphene film on SiC, it requires high-cost substrate materials and the presence of an ultrahigh vacuum environment [33,34]. Other synthesis approaches include the unzipping of carbon nanotubes; solvothermal synthesis (pyrolysis of alcohol and alkali metal for producing GR sheets [35,36]; electron beam irradiation of poly-methyl methacrylate nanofibres, chemical vapour decomposition [CVD], organic synthesis and thermal annealing of GRO etc. [37,38]. Among the various synthesis methods, the CVD technique, exfoliation methods (liquid phase and electrochemical) and chemical reduction of GRO are capable of increasing the production of GR [39,40,41,42]. GR produced by the CVD method has been of transparent nature and has a minimum value of sheet resistance. The synergistic effect of GR composites and chemical modification of GR are capable of solving various problems related to storage, processing, handling and large-scale production. Coleman et al. [43] presented a comprehensive review on GR synthesis techniques and their interrelation with its properties. However, the defects in GR increase the resistance and electrochemical and hydrogen storage ability of GR sheets, and it is really a challenge to manufacture a monolayered GR device using a solution-based method [44].

## 3. Supercapacitors

Effective utilization of renewable energy is only possible with the development of high performing, economical and eco-friendly energy storage and conversion systems. Supercapacitors, also called ultracapacitors, have paved the way to one of the latest and most important sustainable energy paradigms, and their performance is directly connected to the properties of their materials. Supercapacitors can be classified in two categories—electric double layer capacitors (EDLCs) and pseudo-capacitors—which are based on the mechanism of charge storage. The former stores the charge electro-statistically and follows the process of reversible adsorption–desorption cycles of electrolyte ions on the active electrode materials. Electrochemical stability is the prime requirement of the active materials, and in addition to this, they should have a higher surface area and no Faradaic reaction should take place in the EDLC electrode. This type of mechanism has its relevance in carbon-based electrodes. Another type of supercapacitor which under-goes reversible Faradaic reactions is the pseudo-capacitor, which utilizes reversible and fast surface or near-surface reactions for charge storage. The range of specific capacity for GR in EDLCs varies from 62.6 F/g to 215 F/g [45]. Although energy density is higher for EDLC, the stability of the charge/discharge cycle is comparatively lower in the case of pseudo-capacitors. As the time taken is higher for the movement of electrons during redox reactions, the response time is more than that of EDLCs. Energy density, *E*, is governed by the following equation and can be optimized by working on cell voltage (*V*) or/and specific capacitance (*C*):$$E=\frac{1}{2}CV^{2}$$

### 3.1. Performance of Supercapacitor

The performance of a supercapacitor is dependent on various parameters, namely, the thickness of the separator, the electrolyte and the electrode characteristics, which includes mechanical stability, porosity, resistance and volume. GR has been emerging as latest material for supercapacitor applications because it has better features than traditional carbon-based materials. More recently, there has been focus on the use of GR as a supercapacitor due to its large value of electrical double-layer capacitance (around 100–200 F/g using aqueous and organic electrolytes). The value of capacitance can be further enhanced in the range of 200 to 550 F/g through a combination of GR with different pseudocapacitive materials, e.g., manganese oxide, ruthenium oxide and polyaniline. The Ragone plot in Figure 2 depicts the energy loss owing to internal dissipation and leakage losses for sufficiently high and low power [46]. An energy density and specific capacitance of 31.9 Wh/kg and 75 F/g, respectively, were observed with ionic liquid electrolytes for GR based supercapacitors [47], while specific capacitances of 99 F/g and 135 F/g were observed in organic and aqueous electrolytes, respectively [48]. Wang et al. [49] developed supercapacitors with the use of GR materials and observed that energy and power density were 28.5 Wh/kg and 10 kW/kg, respectively, with specific capacitance of 205 F/g. Zhao et al. [50] used mathematical modeling to simulate the approximate capacitance value for a virtual supercapacitor cell, which contained carbon nanosheets comprised of 1–7 GR layers as the electrode material, and found capacitance to be 1.49 × 10^4^ F. Different materials, such as metallic hydroxides, transition metallic oxides and electronic conductive polymer materials, have been explored thoroughly for probable asymmetric supercapacitors uses [51]. Among these, Ni(OH)_2_, with a high specific capacitance (2082 F/g), can be optimistically used for supercapacitor applications [52]. Liu et al. [53] presented a supercapacitor with GR-based electrodes having a specific energy density of 85.6 Wh/kg and 136 Wh/kg at ambient temperatures and 80 °C, respectively. Such an energy density range was close to Ni metal hydride battery, but the supercapacitor had the quality of being charged or discharged in seconds or a few minutes. Recently, an asymmetric supercapacitor with GR/RuO_2_ and GR/Ni(OH)_2_ as its negative and positive electrodes, respectively, was developed, and it had a specific capacitance of 153 F/g and an energy density of 48 Wh/kg at 1.5 voltage in an aqueous solution of 1 M KOH [54]. Yan et al. [55] reported an asymmetric supercapacitor in which porous GR was the cathode and Ni(OH)_2_ was the anode. Optimization of this asymmetric supercapacitor could be achieved by cyclically reversing in the voltage range of 0–1.6 V, and it displayed a fascinating energy density of 77.8 Wh/kg and a maximum specific capacitance of 218.4 F/g. Choi et al. [56] demonstrated a high-performance supercapacitor which was built using chemically modified GR in which embossed-chemically modified GR films were used as 3D macroporous electrodes. Fast ionic transport was facilitated within the electrode because of porosity in the GR structure and the large surface area along with the preservation of good electronic conductivity. Pan et al. [57] used GR–SnO_2_ and GR–ZnO composite materials for supercapacitor uses. GR–ZnO composites displayed superior capacitance of 61 F/g and energy density of 4.8 Wh/kg, and this was better than GR–SnO_2_ materials. Mini et al. [58] developed and characterized high-performance supercapacitor electrodes, which were fabricated by electrophoretic deposition of GR, on which electro polymerization of the poly(pyrrole)-layer (PPy) was carried out. The specific capacitance of the electrode was observed to be 1510 F/g with area and volume capacitances of 151 mF/cm^2^ and 151 F/cm^3^, respectively, at 10 mV/s. Alvi et al. [59] synthesized GR-poly-ethylene-di-oxthiophene (PEDOT) nanocomposites as electrode material by the chemical oxidative polymerization process. Investigation into the charging and discharging properties of GR-PEDOT nanocomposites were done in various electrolytic media, and a specific discharge capacitance of 374 F/g was observed, which proved the viability of this material for supercapacitor applications.

### 3.2. Electrically Conductive Polymers

Electrically conductive polymers (ECPs) due to their high pseudo-capacitance, are of tremendous use in supercapacitors. Accordingly, polyaniline (PANI) [48,60], polypyrrole (PPY) [61] polythiophene (PTH), and their derivatives [62] are various materials which are used for supercapacitor electrodes. PANI, with its high capacitive features, ease of operation and low cost, is deemed to be the most-favorable material [63], but its practical usage is severely restricted due to its poor cycling life and stability issues. Carbon-based materials such as mesoporous carbon (MC), activated carbon (AC), and carbon nanotubes (CNTs) are normally more stable, but they have lower capacitance because of the lower active surface in the materials [64]. This led to composite materials consisting of conducting polymers such as PANIs and carbon-based materials such as CNTs being investigated as supercapacitor electrodes. The synergistic effect of the high pseudo-capacitance of the PANIs and the excellent mechanical and conducting properties of CNTs enabled the achievement of high capacitances and improved stability [65]. Zhang et al. [66] fabricated uniform composites with chemically modified GR and PANI through in situ polymerization and observed the comparatively lower value of electric double-layer capacitance up to 80 F/g. Investigation has been done to develop synergistic materials consisting of GR and ECPs with a high pseudocapacitive energy storage, fast ion/electron conductivity and easily accessible surface area. Wang et al. [67] developed a composite consisting of GR and PANI which had a specific capacitance from 147 F/g (pure Graphene) to 233 F/g and with a reasonable stability. Xu et al. [68] worked on a simple technique of preparing PANI nanowire arrays which were vertically aligned on GRO nanosheets. The specific capacitance of hierarchical PANI–GRO at a discharge current of 0.2 A/g was 555 F/g and 227 F/g at 2 A/g. It was worth noting that, even after 2000 consecutive cycles, hierarchical PANI–GRO and pristine PANI had retained 92% and 74% of their initial capacitance, respectively. Ramaprabhu et al. [69] also worked on GR decoration with various metallic oxides (RuO_2_, TiO_2_, and Fe_3_O_4_) and PANI through a chemical technique. Cyclic voltammetry at different sweep rates in 1 m H_2_SO_4_ showed that RuO_2_/GR composite materials possessed a high value of specific capacitance of 220 F/g at 10 A/g in comparison to composites of other metal oxides, namely TiO_2_ and Fe_3_O_4_. This can be explained by better electrical conductivity and the reversible nature of Faradic reactions of RuO_2_. Wu et al. [70] developed hydrous ruthenium oxide (RuO_2_)/GR sheet composites (ROGSCs) with various loadings of Ru by combining sol-gel and low-temperature annealing processes. Such proposed composite-based supercapacitors exhibited high electrochemical stability (∼97.9% retention after 1000 cycles), increased rate capability, specific capacitance (∼570 F/g) for 38.3 wt% Ru loading and excellent energy density (20.1 Wh/kg) at low operation rates (100 mA/g). Yan et al. [71] investigated a supercapacitor made up of composite consisting of Fe_3_O_4_ and reduced graphene oxide (r-GRO) and had specific capacitance of 480 F/g at a discharge current density of 5 A/g; the corresponding power and energy density were 5.5 kW/kg and 67 Wh/kg respectively. Qu et al. [72] prepared two-dimensional sandwich-like sheets in which iron oxide was grown on a GR surface as high energy anode material for supercapacitors from the direct growth of FeOOH nanorods on a GR surface, which was further converted from FeOOH to Fe_3_O_4_. Such composites exhibited high capacitance (326 F/g), excellent energy density (85 Wh/kg), high-power, and good cycling performance in 1 mol/L LiOH solution. Li et al. [73] found that CeO_2_-GR nanosheet composites, had higher specific capacitance of 208 F/g. This can be explained because of synergistic effect which contributed to the improved electronic conductivity of CeO_2_ and also to the better utilization of GR. Yoo et al. [74] fabricated the electrode and designed a 2D plane utilizing the high surface area of GR. Synthesis of r-GRO CVD film was done, which was tested in solid electrolyte, polymer-gel (PVA-H_3_PO_4_). This electrode had excellent cycle stability up to 3000 charge/discharge cycles and presented specific capacitance around 250 F/g. A nitrogen plasma treatment process was used to prepare nitrogen-doped graphene (N-graphene), which had better capability as an electrode than pure GR. Jeong et al. [75] used the nitrogen plasma process for doping nitrogen into the GR basal planes. The electrode reported a specific capacitance of 282 F/g, which was nearly four times the capacitance of pristine graphene (68 F/g). Basal plane of GR was modified by replacing carbon atoms with nitrogen atoms, and this N-graphene-based super capacitor was capable of working up to 100,000 cycles with 99.8% of capacitance. Energy and power densities have been achieved up to ~48 Wh/kg and ~8 × 10^2^ kW/kg, respectively, in 1 M tetra ethyl ammonium tetra fluoroborate (TEA BF_4_). Qiu et al. [76] reduced the GRO with hydrazine, and subsequently annealing was performed in ammonia atmosphere to produce N-graphene. A maximum capacitance value of 144.9 F/g at a current density of 0.5 A/g was noticed for N-graphene in traditional organic solvent-based electrolyte. Lee et al. [77] prepared N-doped GR via exfoliated graphite oxide which was found to exhibit significantly high oxygen reduction activity. High electrical conductivity and surface area, a large number of edge sites and a pyridinic N site in rGS contributed to oxygen reduction reaction activity. Sun et al. [78] synthesized N-doped GR sheets with nitrogen levels as high as 10.13 atom% via a simple hydrothermal reaction of GRO and urea for high performance supercapacitors. Such sheets exhibited high capacitive behaviors (326 F/g, 0.2 A/g), excellent cycling stability and coulombic efficiency (99.58%) after 2000 cycles. Wen et al. [79] developed a reliable route for preparing highly crumpled N-doped GR nanosheets with ultrahigh pore volume. These nanosheets were found to act as a promising electrode material for supercapacitors with excellent rate capability and high capacity with long-term stability.

Given the electrochemically instable nature of GRO, composites consisting of GRO-PANI are unable to advance beyond the maximum potential of GRO, which is suitable for applications in supercapacitor electrodes. Only a limited quantity of GRO-PANI insulation has been utilized in these composites, because an excess amount of GRO will decrease electrode conductivity [67]. Thus, graphene nanosheets (GNS) have been found to be more favorable than GRO for doping into PANI composites. Zhang et al. [66] fabricated nanofiber composites consisting of GR and PANI through an in situ polymerization in the presence of GRO under acidic conditions, and then hydrazine was used for reducing GRO to graphene. Further, reoxidation and reprotonation were carried out for producing GR-PANI composites. The highest specific capacitance, 480 F/g, was observed for composites containing 80 wt% of GRO at a current density of 0.1 A/g. Even with a current density of 1 A/g, a value of specific capacitance above 200 F/g was achieved. Despite the good cycling stability of the GR-PANI composite, the performance in terms of capacitance was not satisfactory. Thus, Wang et al. [80] innovated a new three-step synthesis technique based on in situ polymerization followed by a reduction–dedoping and then redoping process for preparing a GR-PANI-based electrode for supercapacitors. It had been observed that the rGR sheets were completely and effectively covered by nanostructured PANI granules. It was found that perfect coverage of PANI on GR takes full advantage of the large specific area of GR and could be advantageous for enhancing the electrochemical properties. Such supercapacitors provided specific capacitance of 1126 F/g with a retention life of 84 percent after 1000 cycles. A change to GR from GRO lead to improvement in the mechanical characteristics and caused a higher retention life for GR-PANI composites and higher power and energy density values of 136 kW/kg and 34.8 W h/kg were achieved. Further, Yan et al. [81] carried out synthesis of a GR-PANI composite by providing active sites for the nucleation of PANI and for excellent electron transfer. Synergy of PANI and GR was achieved, as GR sheets provided highly conductive support materials and their larger surface area was well suited for the deposition of nanoscale PANI particles. This resulted in an increase in the maximum specific capacitance of 1046 F/g at a scan rate of 1 mV/s for GNS–PANI composite.

### 3.3. Portable Electronic Devices

Various portable electronic devices deploy flexible supercapacitors for multifarious applications. Wang et al. [82] attempted to develop a GR–PANI composite paper as a flexible electrode by blending the merits of PANI conducting polymer (large capacitance) and of GR paper (mechanical strength, high conductivity and flexibility). There were several important features of this flexible graphene–PANI composite electrode, such as uniform ion accessibility, better electrical conductivity in PANI because of GR paper, and the presence of a triple super-capacitive storage mechanism. Due to these factors, a high tensile strength of 12.6 MPa and a large, stable electrochemical capacitance of 135 F/cm^3^ and 233 F/g for volumetric and gravimetric capacitances, respectively, was shown by GR-PANI electrode. Wang et al. [80] synthesized a flexible GR/PANI hybrid material as a supercapacitor electrode through an in situ polymerization–reduction and dedoping–redoping process. This product was first prepared in an (CH_2_OH)_2_ medium, then treated with hot NaOH solution to obtain the reduced GRO oxide/PANI hybrid material. The characterized results proved better electrochemical properties in comparison to the pure individual components with a specific capacitance of 1126 F/g and a retention life of 84% after 1000 cycles.

### 3.4. Hybrid Supercapacitors

Lithium-ion hybrid supercapacitors (LiHSs), or alternatively named Li-ion capacitors (LICs), are made with a capacitor-type electrode and a lithium-ion-battery-type electrode by using a Li-salt-containing electrolyte [83]. The most commonly used electrode materials in LIHS systems include metal oxides, carbonaceous materials, hydroxides, and intercalation compounds. The real issue in the fabrication of high-performance LIHSs is providing the proper combination of negative and positive electrode materials in the devices. In various aqueous LIHS systems, activated carbon (AC) has been used as a negative electrode with different metal oxides electrodes, such as MnO_2_ [84]_,_ NiO [85], Fe_3_O_4_ [86] and V_2_O_5_ [87]. Out of these, MnO_2_ has been studied by various researchers due to its several advantages of low cost, high specific capacity (1370 F/g), non-toxicity, natural abundance and ease of preparation [88]. Yuan et al. [84] manufactured nano needles (20–100 nm) of a MnO_2_/carbon composite through a process of solid-state grinding. These composites were used as positive-electrode material and this hybrid supercapacitor (Figure 3) reported the highest energy density of 50 Wh/kg for an aqueous system. However, an energy density of 15–30 Wh/kg was found for other metal oxide-based hybrid supercapacitor aqueous systems. Yu et al. [89] utilized a conductive wrapping technique that substantially increased the supercapacitor performance of GR/MnO_2_ nanostructures with carbon nanotubes or conducting polymer by ∼20% and ∼45%, respectively, and a high specific capacitance of ∼380 F/g was achieved. However, these values for supercapacitors were less than those of lithium-ion batteries, which restricted their use to high-power applications such as portable power tools and hybrid vehicles. There a new supercapacitor has been developed at Nanotek Instruments, and it has been claimed that it is capable of storing as much energy per unit mass as nickel metal hydride batteries and can be recharged in seconds [90]. The GR-based super capacitor has been fabricated by mixing GR with an acetylene black called Super P. It is claimed that the battery has an energy density of 85.6 Wh/kg and 136 Wh/kg at room temperature and at 80 °C, respectively, and is close to Ni-metal hydride batteries. GR has shown fantastic electric properties for use as a supercapacitor, and further, the synergistic effect of GR with materials such as conducting polymer, metal oxide and activated carbon can be explored for fabricating supercapacitors and different green energy devices [91]. The need of the hour is optimum utilization of the intrinsic properties of GR, such as its excellent conductivity, large surface area and synergy with different materials for developing energy storage devices. Table 2 shows the use of GR-based materials for capacitor applications.

## 4. Batteries

Researchers have attempted to develop electrode materials and transparent conductors for rechargeable lithium-ion batteries by combining GR and CNTs [93,94]. The presence of CNTs is capable of bridging the defects of electron transfer and also increases basal spacing between GR sheets, which causes substantial property enhancement. Ragone plot clearly shows the dilemma in the selection of power and energy density as depicted in Figure 4. Supercapacitors having high power density are capable of releasing energy in short interval of time but have low energy density. While batteries are proficient in storing high quantities of energy, quick release of energy is impossible due to lower power density, and this is the major challenge for current batteries. There is a dire need of batteries which have high energy and power densities for use in modern smart devices and hybrid vehicles. GR is one of the rare materials which is capable of providing a blend of battery and supercapacitor properties [95].

### 4.1. Li-ion Battery

Lithium-ion-based batteries (LIB) were presented to the market by Sony in 1991 due to research conducted by Mizushima et al. [96], and this has been the most widespread battery technology. Such batteries are the core element of all hand-held devices because of their clean and renewable nature [97]. Generally, graphite is used as the anode material in Li-ion batteries due to its better capacity, higher energy density and increased durability [37]. It is an established fact that graphitic carbon can make LiC_6_ structures, and the low density of lithium in graphite causes relatively low specific capacity in graphite (372 mAh/g) [98]. This can be increased to 744 mAh/g by storing Li on both sides of the GR sheet, which creates LiC_3_ structures [99]. Graphite-based electrodes face disadvantages due to the large lateral size of graphite and due to the long diffusion pathways of lithium into the material. This problem can be reduced by minimizing the lateral (x–y axes) dimension, so that Li^+^ can diffuse into the interlayer space with ease and higher reversibility [100]. The high chemical diffusivity of Li, which is in the range of ∼10^−7^–10^−6^ cm^2^/s, on a GR plane is useful in high-power applications.

#### 4.1.1. Optimization

In addition to the optimization of battery structure and package materials, there is a need to develop high energy density cathodes and anodes. A GR-based anode has come out as a favourable alternative among carbonaceous materials in Li-ion batteries due to higher surface area and better electrical conductivity and chemical tolerance than graphitic carbon [42,101]. Different materials like Sn, Si and transition metal oxides, etc., have been explored as anode materials, and it was found that Si and Li ions can form Li_4.4_Si. This compound has a charge capacity value up to 4200 mAh/g with a small discharge voltage, but the charge volume effect is its main limitation. Silicon and lithium form Li_3.75_Si during the discharge process, which results in 270% increase in Si volume, but also in poor circulation stability [102]. The high capacity of Si anodes in the range of 1 × 10^3^ to 4 × 10^3^ mAh/g makes them quite an attractive proposition [103], but their actual performance is much lower than expected due to fast capacity fading even at low current rates. Likewise, Sn and its oxides, such as SnO_2_, have been studied as anode materials for lithium-ion battery by different researchers. Paek et al. [104] formulated GR nanosheets, which were decked with SnO_2_ nanoparticles by dispersion of reduced GR nanosheets in the ethylene glycol and were reassembled in the presence of SnO_2_ nanoparticles. GR/SnO_2_ had a reversible capacity of 810 mAh/g, and there was significant improvement in the cycling performance in comparison to bare SnO_2_ nanoparticle. However, its real-time use is constrained by a defect, as chemical reduction causes electronic barriers to Li^+^ repulsion, and there are high-volume changes due to lithiation and delithiation reactions [105]. Zhang et al. [106] prepared a three-dimensional composite material consisting of GR and SnO_2_ which had comparatively higher loading of SnO_2_ (89.51 wt%). This composite had high cycling stability with a good reversible specific capacity. There was 97% and 95% capacity retention of 1096 mA h/g and 1073 mAh/g after 150 and 500 cycles, respectively, at a current density of 1 A/g.

#### 4.1.2. Capacity

Despite having a high theoretical capacity of Co_3_O_4_ which is nearly 890 mAh/g, substantial volume expansion is caused by charging and discharging. GR can be used to improve the Co_3_O_4_ electrochemical properties [107,108]. Mn_3_O_4_ with a 936 mAh/g theoretical capacity has the disadvantage of lower electrical conductivity (about 10^−7^–10^−8^ S/cm), and this causes the doping with Co to reach up to a maximum value of 400 mAh/g [109]. CuO has higher catalytic activity and low band gap energy, but in applications as an anode material, it has lower conductive performance and a substantial volume expansion effect. Such limitations can be overcome by developing CuOGR composites [110]. Fe_3_O_4_/GR composites were found to have superb high-rate performance, but not when fabricated by the gas–liquid interface method [111]. Guo et al. [112] introduced a flexible integrated electrode based on layer-by-layer packed GR sheets entrapped by SnO_2_-Co_3_O_4_ nanocubes. The obtained electrode showed excellent reversible capacity of 1665 mA h/g after 100 cycles at 100 mA/g. Wang et al. [113] developed an Fe_2_O_3_/GR composite material using a hydrothermal method, in which inclusion of GR hindered Fe_2_O_3_ from aggregation and also buffered material’s volume expansion. TiO_2_ with GR has the benefit that the oxygen-containing groups on the GR sheets can be reduced after heat treatment [114]. Wang et al. [115] demonstrated a self-assembled TiO_2_-GR hybrid nanostructure for enhancing the high-rate performance of electro-chemically active materials. Liu et al. [116] developed a cathode by embedding the GR sheet approximately 4 ≈ 10 layers thick within vanadium pentoxide. This material exhibited excellent electrochemical performance with higher capacity and longer cycling life. He et al. [117] prepared 3D hierarchically structured aerogels constructed with ultrathin layered nano MoS_2_/GR sheets using a one-pot hydrothermal method for highly efficient lightweight LIBs. The obtained 3D MoS_2_/GR aerogel anode exhibited a capacity of ∼870 mAh/g at 1 A/g after 200 cycles. Ren et al. [118] fabricated an SnS_2_ nanoflake-anchored 3D-GR foam as a flexible Li-ion battery anode using a single-mode microwave hydrothermal method. This exhibited a high reversible specific capacity of 1386.7 mAh/g at 0.1 A/g and a long cycling life with a high capacity of 818.4 mAh/g after 500 cycles at 1 A/g. Chen and Wang [119] fabricated nano GR sheets-wrapped SnCo alloy nanoparticles as an anode material for Li-ion batteries using a chemical reduction approach in an ice water bath. It demonstrated a distinguished higher-than-theoretical capacity of 1117 mAh/g at 72 mA/g, higher than bare nano GR sheets (727 mAh/g) or Sn-Co particles (599 mAh/g). Improvement in cycling performance was achieved due to the complimentary effect of these elements.

#### 4.1.3. Terminals

In comparison to high energy density anodes, cathodes normally have lower energy densities because of their reduced capacity and low voltage plateaus, for example, LiFePO_4_ cathodes. However, recently, GR has been used as an electron-conducting supplement for LIB cathode materials, which is an innovation in energy storage applications [120]. Until now, the use of LIB was limited to different electronic products, but now their use is extended to the domain of electric vehicles. However, there is still a question mark in regard to using this technology to meet customers’ expectations, as a lot of improvement is required in the areas of rate capability, charge capacity and cyclability of LIBs. Kucinskis et al. [120] examined the scope of implementation of GR into lithium-ion battery cathodes for fulfilling such requirements. LiCoO_2_, LiMn_2_O_4_ and LiFePO_4_ have normally been the electrode materials used in LIBs. Li_3_V_2_(PO_4_) is another material which can be explored for use in cathodes, and all of these materials weaken the cathode rate capability. Their electrochemical properties can be improved by the addition of electron-conducting additives. Cu and Li [121] fabricated nano GR sheets and carbon nanotube co-modified Li_3_V_2_(PO_4_)/C composites using a hydrothermal-assisted sol-gel technique. This composite exhibited longer cycle stability and higher rate capability, which had a primary discharge capacity of 147.5 mA h/g at 20 C in the voltage range of 3 to 4.8 volts with 82.7% capacity retention after 2000 cycles. Further, Jeong et al. [122] presented a Li_4_Ti_5_O_12_/N-doped r-GRO composite using dual functional N-doping source for improving the performance rate of Li-ion batteries. The Li_4_Ti_5_O_12_/N-doped r-GRO composite demonstrated good electrochemical performance, cycle stability and specific capacity with a low resistance of 48.4 Ω.

#### 4.1.4. Advancements

Presently, no published studies are available on LiCoO_2_/GR composites. LiFePO_4_, a material having an olivine-type structure with superior cyclability and low manufacturing cost, can be considered an upcoming commercial cathode material [123,124]. This has already been implemented in certain batteries which are used in the automobile sector. [125]. The conductivity of lithium ions as well as electrons is low in LiFePO_4_, and efforts are being made for the improvement of its rate capability through the addition of electrically conductive materials and the modification of grain size and shape [126,127]. Yang et al. [128] fabricated a 3D porous self-assembled LiFePO4/GR composite with the use of a simple template-free sol-gel technique. Electrical conductivity was highly increased by the incorporation of GR nanosheets in a porous hierarchical network and efficient use of the LiFePO4. The obtained composite has a reversible capacity of 146 mAh/g at 17 mA/g after 100 cycles, which is more than 1.4 times higher than that of porous LiFePO4 (104 mAh/g). Dhindsa et al. [129] also synthesized LiFePO4/GR in the presence of dispersed GRO mixed with LiFePO4 precursors and found a six-times increase in the electrical conductivity of the composite as compared to pure LiFePO4 synthesized following otherwise the same procedure without the addition of GR. Moreover, LiFePO4/GR composite exhibited high-rate capability up to 27 C and superb charge–discharge cycle stability for 500 stable cycles as compared to pure. Tao et al. [130] synthesized a new olivine LiFePO4/r-GRO nanocomposite using a solvothermal method. The experimental findings of LiFePO4/r-GRO showed their outstanding electrochemical performance with higher rate capability and better cyclability in contrast to traditional LiFePO4/C nanocomposites. There was a negligible drop in capacity after 200 cycles at a current of 10 C, and a discharge capacity of 119 mAh/g could be delivered at 20 C, pointing out that the electronic conductivity of electrode particles also contributed to achieve a high-rate performance. Zhou et al. [131] also used the same process coupled with subsequent calcination to synthesise a 3D porous composite microsphere of LiFePO4 and N-doped GR. Excellent values of cycle stability, specific charge capacity and rate capability were the basis for the selection of this material in Li-ion batteries. Du et al. [132] developed a LiFePO_4_/GR nano composite using a one-pot in situ hydrothermal method. Measurements showed the discharge capacity of 115 mA/g at 10 C. Phosphate-based Li_3_V_2_(PO_4_)_3_ material is also being studied for using as a cathode in Li- ion batteries. In contrast to LiFePO_4_, this material has a monoclinic structure and has high rate capability with a higher operating voltage, both of which are beneficial when using it as a cathode material [133]. LiMn_2_O_4_ forms a spinel structure, with lithium placed at tetrahedral sites and manganese occupying octahedral sites [134]. Production cost is comparatively on the lower side, but it has less charge capacity in comparison to commonly used cathode materials [135]. Dissolution of Mn^2+^ into electrolyte can create problem, so electrochemical performance can be improved by the addition of other transition metals (most often Ni, Fe and Co) to LiMn_2_O_4_. Jiang et al. [136] used nano GR sheets as a planar conductive additive in spinel LiMn_2_O_4_-based electrodes for increasing the electronic conductivity of LiMn_2_O_4_. The specific capacity and cycling performance of LiMn2O_4_ were increased when nano GR sheets and acetylene black were present together at an appropriate weight ratio in the LiMn_2_O_4_-based electrode. Moreover, in contrast, when nano GR sheets were present in large quantity, there was no improvement or decrease to the rate performance and conductivity of LiMn_2_O_4_. Bak et al. [137] developed a LiMn_2_O_4_/GR (27 wt%) cathode material which was nanocrystalline and had superb discharge capacity and high rate capability. Moreover, it had a 90% and 96% discharge capacity at 1 C and 10 C, respectively, for 100 cycles. It was claimed that the material had superb electrochemical properties to be connected with a good dispersion of the reduced GRO nano sheet template. Pyun and Park [138] fabricated the composites of LiMn_2_O_4_ nanoparticles in a GR matrix to compensate for the low electronic conductivity of the LiMn_2_O_4_ cathode. The GR/LiMn_2_O_4_ electrode showed increased discharge capacity and rate capability over a pristine LiMn_2_O_4_ electrode. This may be attributed to the higher surface area of LiMn_2_O_4_ nanoparticles and superior electronic conductivity with the presence of GR. The composites of LiMn_2_O_4_ nanoparticles with GR were also effective in stabilizing the cyclic performance of the LiMn_2_O_4_ nanoparticle cathode. Wang et al. [139] prepared LiMn_1-x_Fe_x_PO_4_ nanorods on reduced GRO sheets, finding that LiMn_0.75_Fe_0.25_PO_4_ (x = 0.25) had good discharge capacity even at a higher discharge of 100 C. It was demonstrated in experiments that slightly oxidized GR sheets showed a unique substrate for the growth of nanocrystals into well-defined morphologies. It has been found that the blend of nanoparticles, doping and improved GRO preparation have provided excellent rate capability, which is difficult to achieve with other techniques [140,141], and a discharge rate of 65 mAh/g was obtained at 100 C. Zhu et al. [142] fabricated and tested 3D macroporous GR-based Li_2_FeSiO_4_ composites as the cathode materials for Li-ion batteries. Their performance was compared with the performances of 2D nano GR sheet-based Li_2_FeSiO_4_ composites and Li_2_FeSiO_4_ composites without GR. When compared with the 2D-Li_2_FeSiO_4_ composites, 3D macroporous GR-based Li_2_FeSiO_4_ composites exhibited better performances, with discharge capacities reaching 313 mAh/g at 0.1 C and 108 mAh/g at 20 C. Three-dimensional macroporous GR provided higher surface area than 2D GR sheets for Li_2_FeSiO_4_ and also allowed the electrolyte ions to diffuse inside and through the structure of the cathode material.

The application of GR as an electrode material is still in the nascent stage, and further research must be carried out in order to understand the electrochemical processes of GR-based electrodes, and their application aspect in Li-ion batteries must be explored. Table 3 depicts a summary of GR-based cathode and anode materials in LIBs.

### 4.2. Sodium and Calcium Ion Battery

Rechargeable LIBs are the backbone of portable electronic devices and are also commonly used in small vehicles and different power devices. These batteries exhibit excellent performance and have higher energy density in comparison to other rechargeable batteries [143]. The main concern is that lithium is a rare element among light metals, and its concentration is approximately 35 ppm in the upper continental crust [144]. For this reason, available resources of Li on Earth cannot fulfil the ever-growing demand of LIBs, and other alternatives have to be explored. This has led to research on sodium-ion batteries (NIBs) and calcium-ion batteries (CIBs) [145,146]. Recent density functional theory (DFT) studies anticipated that the presence of defects would increase Li adsorption on GR, which would yield higher gravimetric capacity [147]. The influence of defects in GR on the adsorption of Na and Ca still needs to be investigated. Datta et al. [148] worked in this direction and performed first-principles calculations based on DFT for investigating the Na and Ca adsorption on GR with different percentages of Stone-Wales (SW) and divacancy defects. The reported merits of NIBs and CIBs included natural abundance, economical, low reduction potential, chemical safety and lower mass-to-charge ratio. Moreover, nature does not store energy with Li ions, but rather with Na and Ca ions [149]. Table 4 shows the quantum of the charge transfer for different positions.

#### 4.2.1. Background

One newly founded company named ‘Aquion energy’, based in Pennsylvania in 2014, had commercially available NIBs (using a manganese oxide–spinel structure host as the cathode and NaTi_2_ (PO4)_3_/activated carbon as the anode) with a cost/kWh capacity comparable to that of Ni-Fe batteries which could be used as a backup power source in electricity micro-grids and which operate independently of the centralized grid. Another UK-based company, ‘Faradion’, claimed that they had already developed different NIB materials with greater energy densities than the common Li-ion material (i.e., LiFePO_4_), and this dispelled the doubt that sodium-ion materials are not able to attain high energy densities. OXIS energy, based on the Culham Science Centre (Abingdon, UK) in Oxfordshire, offered a good energy density of 300 Wh/kg of NIBs in 2014. Yabuuchi et al. [150] presented a comprehensive review on the energy density of NIBs. Ding et al. [151] carefully designed a 3D macroporous network of carbon nanosheets with a very broad GR interfloor distance (0.388 nm) which was able to facilitate high-quality Na intercalation even below 0.2 V vs. Na/Na^+^. Wang et al. [152] showed that a dual-GR rechargeable Na battery manufactured with the use of exfoliated GR as both the negative and positive electrodes provides the maximum operating voltage among all Na ion full cells reported to date, along with the highest energy density at 250 Wh/kg. Wu et al. [153] disclosed that GR coated with antimony sulphide, which was manufactured using a solution-based synthesis method, can be used as an anode material for NIBs. Excellent cycle performance and a superb rate capacity of 730 mAh/g was achieved by this technique. Excellent performance was achieved due to fast diffusion of Na ions from the nanoparticles and electrical transport from the close contact between the active material and GR.

Phosphorous element (P) is perceived as an excellent anode material for NIBs because of its exceptionally high theoretical capacity of 2596 mAh/g. However, the low conductivity of phosphorous and fast structural degradation, which is caused by extraordinary volume expansion (>490%) during cycling, are the major drawbacks. Zhang et al. [154] developed a new design for anode structure with a new technique to manufacture flexible paper composed of nitrogen-doped GR and amorphous phosphorus, which was capable of resolving this problem. Doborta et al. [155] performed a DFT study on Na interaction with doped GR, both in non-oxidized and oxidized forms. It has been suggested that the oxidation level of doped graphene-based materials should be carefully controlled during its use as sodium battery electrode material, as the optimal oxidation level depends on the dopant type. GR-based electrodes, because of their highly anisotropic morphology, develop a compact uniaxially oriented stacked structure. Yun et al. [156] demonstrated that the self-standing electrodes formed of crumpled GR nanosheets are capable of substantially increasing the power capability of GR-based anodes in NIBs. Such electrodes are capable of delivering a power density of nearly 20,000 W kg^−1^, which is higher than the Li storage capacity of conventional GR paper electrodes. A phosphorene-GR hybrid material to be used as a high-capacity anode for Na-ion batteries had been presented by Sun et al. [157] which showed a specific capacity of 2440 mAh/g at a current density of 0.05 A/g and an 83% capacity retention after 100 cycles. Xu et al. [158] fabricated three-dimensional N-doped GR foams to act as an anode for substantially increasing the overall performance of NIBs. It has been found that the prepared foams produced a large initial reversible capacity of 852.6 mAh/g at a current density of 500 mA/g; however, after 150 cycles, the foam was able to achieve a charge capacity of 594 mAh/g, retaining 69.7% of the initial charge capacity, and this performance was better in comparison to that of other carbonaceous materials. Xie et al. [159] conducted comparative analysis of SnO_2_/N-doped GR nanohybrids and SnO_2_/GR as anode materials for NIBs. The findings manifested that the N-doping caused improved electron transfer efficiency of SnO_2_/N-doped GR as compared to SnO_2_/GR counterpart. Qu et al. [160] fabricated a SnS_2_-r-GRO composite with superb electrochemical performance for anode of NIBs. Such electrode showed high rate performance of 544 mAh/g at 2 A/g, a high charge-specific capacity of 630 mAh/g at 0.2 A/g and a long cycle-life of 500 mAh/g at 1 A/g for 400 cycles. This may be due to the fact that the higher interlayer spacing in the SnS_2_ layered structure is capable of accommodating volume change in Na-Sn insertion and de-insertions.

#### 4.2.2. Performance Characteristics

Performance characteristics of batteries, such as their specific capacity and operating voltages, are highly dependent on the electrochemical properties of the electrode materials [146], and therefore, proper selection of the electrode materials for NIB and CIB technologies is a major task. This can be accomplished by investigating the chemistry and structure of electrode materials, which perform well for Li intercalation. Graphite has been well-accepted as an anode material in LIBs, but the lower value of gravimetric capacity even in NIB and CIB requires other materials to be thought of. Medeiros et al. [161] and Stournara et al. [162] observed that materials like GR and graphene oxide can make suitable replacements for graphite in LIBs. Liu et al. [163] noticed that poor binding of magnesium and sodium on GR layers can be due to the comparative quantities of ionization energy of the metal atoms and coupling between GR and the metal cations. Therefore, two probable methods for the electrode design of NIBs and CIBs based on layers of carbonaceous materials are, first, to choose materials which bond strongly to metals and second, to choose materials that have more inter-layer spacing than graphite. Niaei et al. [164] addressed the issue of weak binding and applied density functional theory to demonstrate the effectiveness of hydrogenation for both calcium and sodium ion batteries. It has been observed that hydrogenated defective GR can be used as an anode material for improving the performance of rechargeable batteries in comparison to GR, using metals that are lower in cost than Li. Further, Niaei et al. [165] used computational methods to demonstrate that GR nanoribbons bind Ca and Na with higher strength than GR sheets. Further increase in binding strength was carried out by functionalizing the edge of the nanoribbon with oxygen-containing groups. David et al. [166] examined the synthesis along with mechanical and electrochemical performance of layered free-standing papers fabricated of r-GRO flakes and acid-exfoliated few-layer MoS_2_ for using as electrode in NIBs. This electrode indicated excellent Na cycling ability and charge capacity of 230 mAh/g. NIBs and CIBs are capable of being explored as batteries for energy storage and hybrid vehicles, where the prime requirements are excellent power and low-cost, respectively. If such batteries have a high cycle life and are safe in operation, then both sodium and calcium ion batteries occupy an important place in the rechargeable battery market share along with high-energy lithium systems.

## 5. Fuel Cells

The design and development of proficient energy storage and conversion devices is mandatory for exploring the use of renewable energy sources in an effective manner at all levels. These systems need to be low cost, high performing and environmentally friendly, and efforts are being made all across the globe to accomplish these goals [167,168]. Fuel cells have been considered the most reliable energy storage and conversion electrochemical system since their development way back in 1893. But major thrust was received in the 1980s, when importance was given to fuel cells for their applications in producing electricity and heat without producing any toxic or pollutant by-products [169].

### 5.1. Design and Development

The basic design of fuel cell systems consists of hydrogen and oxygen, and these are divided by proton exchange membrane. This proton conductor membrane in fuel cells must exhibit excellent proton conductivity, hydrolytic and thermal stability, chemical and electrochemical stability. In addition to this, it must be low in cost, should have excellent mechanical strength, excellent water uptakes, low permeability to reactant species, good chemical properties and must be suitable for wide variety of fuels. Basically, the membrane performs two important functions in a fuel cell: first of all, it performs the function of electrolyte between the anode and cathode for ionic conduction, and second, it works as a separator that keeps the two reactant gases apart [170]. Rapid progress in membrane technology has a direct outcome on fuel cell performance. Advancement in material technology has a direct linkage with membrane performance and innovation in excellent energy storage and conversion devices. There are a few types of membranes available in the market, such as Nafion (DuPont de Nemours, Wilmington, DE, USA), perflourinated ionomer (PFI) membranes and Dow membranes (Dow Chemical, Midland, MI, USA). Nafion is lower in cost than Dow membranes but is low performing [171]. Cao et al. [172] presented a new approach for preparing polyethylene oxide (PEO)/GRO composite-based membrane without any chemical modification for low temperature polymer electrolyte membrane fuel cells. PEO/GRO composite membrane had a tensile strength of 52.22 MPa and Young’s modulus 3.21 GPa. There was an increase in ionic conductivity from 0.086 to 0.134 S cm^−1^ with an increase in temperature from 25 °C to 60 °C for this membrane. Lee et al. [173] obtained Pt nanoparticles via in situ Pt nanoparticle deposition onto GRO using a microwave method, finally resulting in Pt-graphene (Pt-GR). There was a significant increase in cell performance when membrane electrode assemblies were fabricated with the Nafion/GRO. However, there was not sufficient enhancement in cell performance at some values of relative humidity due to the low water retention ability of GR. Different Nafion/xPt–G/ySiO_2_ composite membranes (x values: 0.5, 1.5 and 3.0 wt% and y values: 0, 1.5 and 3.0 wt%, respectively), were manufactured for which water uptake and proton conductivity depicted the same behaviours [174]. It was found that for less than 1.5 wt% in Pt-GR content, there was an increase in cell performance with SiO_2_ due to the good retention ability of SiO_2_ for water produced from the Pt site on GR. However, there was a decrease in cell performance upon increasing Pt-GR content above 1.5 wt%, which was attributed to blocking the impact of proton conduction because of excessive inorganic filler and electron loss through the Pt network. Yang et al. [175] successfully fabricated various Nafion/xPt–TiO_2_/(1 − x) GRO composite membranes. Proton conductivity and ion exchange capacity were found to be improved up to a certain content of GRO, and excessive concentration caused blocking effect and reduced performance. The experimental findings indicated that there was appreciable in-cell performance due to synergistic action caused by the addition of certain amounts of Pt–TiO_2_ and GRO to Nafion. Lee et al. [176] enhanced the moisture retention and proton conductivity by incorporating a combination of constituent materials into SPEEK membranes in order to prepare new, self-humidifying composite membranes (SHMs) for proton exchange membrane fuel cells. SHMs were therefore prepared with the inclusion of carboxyl-functionalized graphene (G(c)) and phosphotungstic acid (PWA) with varying proportions into the SPEEK film. The results confirmed improvement of the self-humidifying properties at temperatures above 60 °C by collective inclusion of G(c) and PWA within SPEEK and the new SHMs have potential for use in medium-temperature DMFCs.

### 5.2. Properties Based Applications

Various carbon materials due to their stability, availability in abundance and environmentally friendly nature have excellent scope in energy devices. This scope is further widened due to the superb thermal stability of these materials in different acidities and media over a wide temperature range [177]. Earlier, fuel cells deployed CNTs as, these performed the role of catalysts for improving their performance [178]. With the passage of time, focus has shifted to two-dimensional carbon materials, i.e., GR for use in fuel cells [179]. GR, because of its excellent physical properties, is considered as the most serious contender to resolve the problems related to electrochemical applications of fuel cells [180]. A comprehensive survey has been presented on the latest ground-breaking advancements related to theoretical as well as experimental findings in chemical science and engineering-related GR-based membranes. Aspects of design, manufacturing and applications have been summarized along with the separation performance of GR-based membranes [181].

Recently, researchers have displayed the encouraging effects of incorporating GRO in the proton exchange membrane due to its softness and amphiphilic nature [182]. Moreover, it is possible to enhance proton conductivities with GRO-based membranes because of interactions between the intermolecular H-H bond and the structure of membranes [183]. With the energy crisis and environment pollution becoming serious, polymer electrolyte membrane fuel cells, as an environmentally friendly power source, are attracting much attention. GRO nanosheets are designed to explore the utilization of GR as a prospective filler in order to obtain the required improvements in the polymer electrolyte membrane. Polymer electrolyte membrane fuel cells (PEMFCs) normally operate with Nafion membranes because of their excellent ionic conductivity, superb chemical stability and high mechanical strength, although it is quite expensive. Over last few years, there have been efforts to modify Nafion with the different forms of GR materials. According to Figure 5, GRO is compatible with the majority of the polymers and solvents used to create composite membranes for fuel cell applications.

Ansari et al. [184] inculcated GRO in a polymer matrix for the modification of Nafion and noticed encouraging alteration in ionic domains. A composite membrane developed by Wang et al. [185] included customized GRO in Nafion resin for fuel cell applications. The performance of such a fuel cell having a 3 wt% GRO/Nafion composite membrane was the same as that of the pristine Nafion membrane; however, the composite membrane had better mechanical characteristics than Nafion. Peng et al. [186] carried out modification of Nafion chains by preparing a nano hybrid membrane of Nafion and GRO. First, they obtained nano hybrids of Nafion and GRO, and then they characterized them with Raman spectroscopy, Fourier transform infrared spectroscopy (FTIR) and X-ray photo-electron spectroscopy (XPS) over atom transfer radical addition (ATRA) reaction amid the C=C group of GRO and the C-F group of Nafion. Inclusion of GRO with Nafion chains improved the interfacial compatibility among them. With this, the proton conductivity of GRO-included membrane was increased by 1.6 times in comparison to recast Nafion membrane. Kumar et al. [187] prepared GRO/Nafion composite membranes for PEMFCs. The proton conductivities of Nafion recast membranes, GRO (4 wt%)/Nafion composite and Nafion 212 and at 30 °C and 100% humidity were 0.043, 0.078 and 0.068 S cm^−1^, respectively. Kim et al. [188] demonstrated that the phosphotungstic acid (PW)-mixed GRO–Nafion (Nafion/PW-mGRO) membrane had higher proton conductivity in comparison to pure and recast Nafion membranes. The power density of this hybrid membrane was 841 mW cm^−2^, and pure Nafion membrane had a power density of 210 mW cm^−2^ at 80 °C under 20% RH. Choi et al. [189] reported an advanced chemical mechanism for developing of polymer electrolytes. Such composite membranes had lower methanol crossover and higher proton conductivity.

Further, Zarrin et al. [190] used functionalized GRO (f-GRO) for modifying Nafion composite membranes for PEMFC. Hummers’ method was used for producing GRO and modification of Nafion electrolyte with f-GRO. Functionalization of GRO had taken place with mixture of toluene and 3-mercaptopropyl trimethoxysilane. The addition of 5 and 10 wt% of f-GRO into the Nafion polymer matrix caused an improved water uptake of 2% and 6% and IEC of 0.93 and 0.96 meq/g contrasted to recast Nafion membrane having an IEC value of 0.91 meq/g. Moreover, there was a four-times increase in proton conductivity with f-GRO addition in comparison to simple recast Nafion membrane at 120 °C and 30% RH. The sulfonated polyether ether ketone (SPEEK) has been recognized as a widely used polymer for PEMFC. Modification of SPEEK has been done many times with GR for optimizing membrane performance. Mishra et al. [191] studied the influence of low- and high-degree GRO (HGRO) oxidation on SPEEK-Nafion/GRO nanocomposite membranes. It was confirmed that the oxygen functionality in GRO and HGRO had a concentration of 28.4% and 31.8%, respectively, and that GRO had a larger size than HGRO. Further, polybenzimidazole (PBI) polymer also has good compatibility with GRO, in which recent modification has been undergone with SiO_2_, TiO_2_, and zirconium phosphate (ZrP), and it was found to be useful as a proton exchange membrane (PEM) in PEMFCs [192]. Feng et al. [193] successfully prepared a Nafion/GRO composite membrane for PEMFC, which exhibited better proton conductivity as contrasted to the recast Nafion membrane, specifically in lower humidity environments. There was an enormous rise in proton conductivity due to rearranging of the microstructures of the Nafion matrix. Zarrin et al. [190] replaced the proton exchange membrane by introducing an f-GRO Nafion nanocomposite (f-GRO/Nafion) for high-temperature PEMFC applications. Nano GRO sheets were fabricated from graphite flakes using the modified Hummer’s method, and substantial improvement was exhibited for f-GRO/Nafion membranes (four times) over recast Nafion at 120 °C and 25% humidity. Lim et al. [194] fabricated and characterized the sulfonated poly(ether sulfones) which contained a combination of cis- and trans-mesonaphthobifluorene moiety. The membranes were studied for water uptake, ion exchange capacity (IEC) and proton conductivity. Yang et al. [195] introduced the triazole f-GRO and observed that the PBI/GRO composite membrane had improved tensile strength and proton conductivity in comparison to pristine PBI membrane. It was found that f-GRO can be an excellent alternative for preparing inorganic polymer electrolytes for PEMFCs. In addition to these well-known polymers, GRO has been used for the modification of several other polymers which exhibit the functionalization, flexibility, and substantial contribution of GRO. Xu et al. [183] prepared two types of composite membranes, namely PBI/GRO and PBI/sulfonated GRO for high-temperature PEMFCs. Phosphoric acid was loaded on membranes in order to provide the required proton conductivity. Such membranes exhibited ionic conductivities of 0.027 S cm^−1^ and 0.052 S cm^−1^. A power density of 600 mW cm^−2^ at 175 °C was achieved with PBI/sulfonated GRO membrane. Sharma et al. [196] carried out modification of GRO with silica and homogeneously mixed it into chitosan matrix and polyvinyl alcohol (PVA) for PEMFCs at diverse concentrations of GRO. A substantial increase in the mechanical, chemical and structural properties of the membrane was observed with the addition of GRO; for example, proton conductivity increased from 6.77 × 10^−2^ to 11.2 × 10^−2^ S/cm on addition of GRO in the PVA membrane. Further, there was improvement in the mechanical and thermal stability of PEM with the addition of GRO. Ye et al. [197] described a novel PEM using protic ionic liquids (PILs) with ionic liquid polymer-modified GR sheets [PIL(NTFSI)-G]. Ionic conductivity was increased by 257.4% and tensile strength by 345% on 0.5 wt% loading of GR in PIL (NTFSI). There was a 20% savings in cost with the addition of GR in costly PIL. Sulfonated polyimide (SPI)/sulfonated propylsilane GRO (SPSGRO) was also explored as an upcoming candidate for PEMs. Pandey et al. [198] fabricated this composite membrane for the promotion of internal self-humidification, improvement of water-retaining characteristics and the increase of proton conduction. In single-cell DMFC tests, SPI/SPSGRO-8 indicated 75.06 mW-cm^–2^ maximum power density, which was more than the value exhibited by Nafion membrane (62.40 mW-cm^2^).

### 5.3. Hybrid Fuel Cells

The direct methanol fuel cell (DMFC) is another class of PEMFC and is visualized as the leading substitute for power-generating systems in coming times due to its simple design and operation with high energy performance. Using an aqueous methanol solution as fuel, Figure 6 depicts the fundamental design and operation of a DMFC [199].

Yan et al. [200] placed a single layer GR between two thin films of Nafion in order to fabricate a unique membrane for DMFCs. This was carried out to reduce methanol permeability by retaining good selectivity. The process of adding the GR layer led to the enhancement of proton permeability and the reduction of methanol permeability by 68.6% as compared to a pure Nafion membrane. Apart from Nafion, SPEEK is also a remarkable GRO-compatible polymer for PEMs and has been employed many times with functionalized GRO in DMFCs and delivered superb results in accordance with the desired characteristics of PEM. Recently, Yin et al. [201] reused the SPEEK with GRO sheets for preparing hybrid PEM for DMFCs. GRO sheets were functionalized by histidine molecules and its effects on the fractional free volume, crystalline structure, thermal stability, cross sectional morphology and polymer chain stiffness of the composite membrane were examined. There was an increase in proton conductivity by 30.2% and power density by 80.7% for the hybrid membrane as compared to plain SPEEK. Jiang et al. [202] used sodium dodecylbenzene sulfonate (SDBS)-adsorbed GRO as a filler for the modification of SPEEK which resulted in improving the water uptake, ion-exchange capacity and proton conductivity, but it reduced the methanol permeability through the SPEEK membranes. Due to such excellent features, composite membranes with optimized SDBS-GRO contents delivered excellent performance in DMFCs in comparison to pure SPEEK or Nafion 112 membranes.

Further, inspired by the bio adhesion of mussels, He et al. [203] prepared a nanocomposite membrane using polydopamine-modified GRO (DGRO) sheets bearing –NH_2_ and –NH– groups. Such DGRO sheets were interlinked and homogeneously dispersed in a SPEEK matrix, which makes peculiar rearrangement of the nanophase-separated structure and chain packing of nanocomposite membrane. The maximum values of current density and power density were increased by 47% and 38%, respectively. Chien et al. [204] demonstrated that Nafion, with a low content of sulfonated graphene oxide (SGRO), displayed a peculiar viscosity pattern and showed better SGRO dispersion within the Nafion. Such a membrane showed less methanol and water uptake, a lower swelling ratio, increased proton conductivity and very high methanol selectivity, all of which are useful for implementing in DMFCs. Further, Heo et al. [205] prepared a new type of membrane consisting of SPEEK and SGRO with different contents of SGRO for improving proton conductivity. Such a membrane also improved the mechanical properties and obstructed the flow of water and methanol molecules through it, in addition to enhancing proton conductivity. Such characteristics were advantageous for the selection of SPEEK/SGRO membranes in DMFCs. Beydaghi et al. [206] synthesized nano Fe_3_O_4_/SGRO sheets using a hydrothermal method and blended with a SPEEK/PVA matrix at various concentrations of Fe_3_O_4_/SGRO nanosheets, suggesting its potential application in DMFCs. This inclusion of Fe_3_O_4_/SGRO was helpful in improving proton conductivity, mechanical stability and methanol barrier properties. It was found that 5 wt% content of Fe_3_O_4_/SGRO gave optimal results for tensile strength, proton conductivity, power density and low methanol permeability. Kumar et al. [207] used a flow-directed assembly of GRO solution to make free-standing GRO paper of approximately 100 μm thickness. Electrochemical characterization of such membrane electrode assembly showed proton conductivity in the range of 0.041 S/cm–0.082 S/cm at temperatures of 25–90 °C, with a peak power and current density of 8 mW/cm^2^ and 35 mA/cm^2^, respectively, at 60 °C for the DMFC. The development and performance of DMFC is severely affected by crossover of methanol through the PEM from the anode to cathode. Yuan et al. [208] highlighted the scope of using GRO as a methanol-blocking thin film made with a layer-by-layer assembly of poly(diallyldimethylammonium chloride) on the surface of Nafion membrane in the DMFC. The results demonstrated that such composite membranes reduced the permeability of methanol as compared to pure membrane. Choi et al. [209] exploited the use of Nafion/GRO membrane as electrolyte material for DMFC. The performance of DMFC with this composite membrane significantly improved in severe conditions as compared to using Nafion 112 membrane. Lin and Lu [210] presented GRO-laminated Nafion 115 as a PEM for a DMFC and found this membrane to give better results as compared to GRO-dispersed polymer composite membrane. The methanol permeability of the GRO-laminated membrane exhibited 70% lower methanol permeability as compared to Nafion 115 with 22% decrease in proton conductivity.

#### Microbial Fuel Cell

Presence of biodegradable substrates in wastewater can be utilized for bioelectricity generation by microbial fuel cell (MFC) [211,212]. Khilari et al. [213] developed the GR-modified polyvinyl alcohol silicotungstic acid (PVA-STA) membrane for MVCs by solution casting. Modified Hummer’s method was deployed for synthesizing GRO and included a PVA-STA solution for obtaining a membrane of 100 μm thickness. It was confirmed by electron impedance spectroscopy (EIS) that the conductivity of the PVA-STA-GRO membrane has been improved to 0.065 S/cm by the addition of GR; it was approximately 0.046 S/cm of pure PVA-STA membrane and 0.062 S/cm of Nafion 117 membrane. There was an improvement in tensile strength with the addition of GRO, and values of 31.3, 39.1 and 37 MPa were observed for Nafion 117, PVA-STA-GRO and PVA-STA membranes, respectively. Further, GRO-filled membrane had a superb power density of 1.19 W/m^3^ as compared to the 0.88 W/m^3^ of Nafion 117. Alkaline fuel cells (AFCs) have also been considered in past years for applications in the Apollo space program and space shuttle program. Ye et al. [214] developed a GR-modified polyvinyl alcohol (PVA/GRO) composite membrane as an electrolyte for AFCs. The presence of GR had been found to improve the ionic transport, as it formed well-connected and uninterrupted ionic channels in the membrane structure. Moreover, there was an increase in the ionic conductivity of GRO-filled membranes by approximately 126% and methanol permeability was reduced by nearly 55% with 0.7 wt% of GR. The addition of GRO also caused a substantial increase in tensile strength and power density of 73% and 148%, respectively. Researchers have been working to improve the performance of MFCs by exploring new materials for electrodes. One such method is the modification of anode materials using GR-based materials with or without bridging binders. Binders can be different biological or polymeric materials or metals and metallic oxides. Zhang et al. [215] observed improved electrochemical performance of a MFC by deploying a GR-modified anode, while Huang et al. [216] demonstrated that GRO nanoribbons could increase the extracellular electron shuttling in bio-electrochemical systems. These results demonstrated the ability to use GR in MFC electrodes for performance enhancement. The productivity of MFCs had also been increased by applying various hybrid materials on GR. Although Pt metal is most preferred as a catalyst for the cathode due to its noble characteristics, due to its high cost, work is being done on using Mn and Fe. These alternative metals gave comparable performances, but further examination of their durability is still to be carried out [217]. In 2011, the first attempt was made to integrate GR onto the cathode of a MFC with the use of Nafion as a binder material. Further, in different studies, conductive polymers such as PANI and Nafion were applied as binder materials for GR-based cathodes [218,219]. The average power density of polymer GR-based cathodes was found to be is 742 mW/m^2^, which represented an average increase of 39 times in comparison to the control cathode. Spectacular results were achieved when Nafion was used with N-doped GR cathode, which yielded a power density of 1350 mW/m^2^. Results were comparable to those achieved with Pt cathode. Leong et al. [220] filled SPEEK with a single-layer GRO and performed tests on MFCs using Nafion 117 and GRO-SPEEK membranes. The MFC system with Nafion 117 membrane produced the maximum power density (1013 mW/m^2^) followed by the MFCs with GRO-SPEEK (902 mW/m^2^) and SPEEK (812 mW/m^2^) membranes.

### 5.4. Advancements

In particular, substantial effort has been made toward developing alkaline anion exchange membrane fuel cells (AEMFCs) because of their high energy conversion efficiency, good power density and lower formation of pollutants. The anion exchange membrane (AEM), which acts as an electrolyte to transport anions, is one of the key components of AEMFCs. Wang et al. [221] successfully fabricated and characterized PBI/ionic liquid f-GRO nanocomposite AEMs. The resulting membrane exhibited good alkaline and thermal stability and superb mechanical properties along with high conductivity (more than 0.01 S/cm). Liu et al. [222] prepared different novel composite AEMFCs by incorporation of quaternized GRs (QGRs) into the chloromethylated polysulfone (CMPSU) followed by quaternization and alkalization. It was found that quaternized polysulfone (QPSU) with 0.5% QGRs exhibited four-times enhancement in bicarbonate conductivity as compared to pure QPSU membrane at 80 °C, while there was a three-times improvement in tensile strength and Young’s modulus with the addition of 0.25% QGRs in QPSU. Bayer et al. [223] presented a novel class of alkaline AEM as KOH-modified multilayer GRO paper. The maximum anion conductivity was found to be 6.1 mS/cm at 70 °C, and OH^−^ was confirmed to be the dominant charge carrier by utilizing anion- and proton-conducting blocking layers. The value of the ion exchange capacity as measured by titration was found to be 6.1 mmol/g. Although extensive work has been focused on the synthesis of AEMs, just a few studies have depicted enhanced ionic conductivity with simultaneous suppression of unfavorable mass transport and increased thermal and mechanical properties.

Therefore, GR has delivered encouraging results in numerous fuel cell applications and provided solutions to some key challenges and to other technical issues related to cell membranes. Presently, extensive research is being done on the use of GR for diverse energy-related applications, and it is expected that the remarkable features of this carbonaceous material will help to achieve new heights in energy efficiency. Different properties of GR-based materials can be utilized for developing composite membranes which can be put to use in fuel cells, and this has been summarized in Table 5.

## 6. Solar Cells

With the advent of industrialization, lot of technological and economical activities are being pursued to meet the needs of the increased population, and these create pressure and higher energy demands. This has culminated in the burning of conventional fossil fuels, which has created various environmental concerns such as air pollution, global warming, acid rain and, most importantly, the climate change. Focus has shifted to producing electricity from renewable sources and the utilization of solar energy through use of photovoltaic panels is being seen as a promising alternative [4]. The conversion efficiency of various solar photovoltaic technologies is shown in Figure 7. It is noteworthy that the n/p-type of semiconductor, because of its larger carrier mobilities and direct energy gaps, results in multijunction solar panels with the best efficiency and superior stability when compared to those on the market [224].

### 6.1. Design and Development

New developments are occurring at a fast pace across the globe in the use of photovoltaic (PV) solar energy for meeting energy requirements. Developments are related to use of new materials, design aspects of devices, latest production technologies and improving the efficiency of solar cells. GR, due to its characteristics of extraordinary high carrier mobility and superb electron transfer, is a suitable candidate for use in low-cost and efficient PV devices [10,225]. Among these PV devices, GR/Si Schottky-barrier solar cells are highly attractive because of their simple structure, high-efficiency capabilities and low cost [226]. For Schottky-barrier solar cells, the Schottky barrier height (SBH) is a crucial factor which determines the device performance, and efficient charge separation requires a larger value of SBH. GR is chemically doped for increasing SBH, as chemical doping of GR is the most widely prevalent approach for increasing SBH, determined by taking the difference between the electron affinity of Si and the work function of GR [227]. Li et al. [226] combined highly conductive semi-transparent GR sheets with an n-type silicon (n-Si) wafer for fabricating solar cells with PCE ≤ 1.5% and an illumination intensity of 100 mW/cm^2^. Jiao et al. [228] improved the efficiency of GR/silicon solar cells by more than 100% by deploying an interface tailoring approach for inserting a thin GRO layer. Systematic research was done for checking the influence of the GRO interfacial layer by varying the thickness of the GRO layer and changing the annealing temperature. Li et al. [229] developed Schottky junction solar cells by depositing semi-transparent and highly conductive GR films on n-type Si wafers. It had been observed that GR also acts as a transparent electrode in addition to its contribution to charge separation and transport. However, higher power conversion efficiency was not achieved with this cell because of the weak junction and contact between the interfaces. Further, Zhang et al. [230] and Ye at al. [231] extended the basic idea of Schottky junction solar cells by utilizing semiconducting substrates such as CdSe and CdS instead of Si. In addition, Miao et al. [227] introduced more functional layers, such as trifluoromethane-sulfonyl-amide, as the p-dopant and enhanced the conversion efficiencies. Only limited improvement was obtained for these devices, and further performance enhancement of this solar cell based on GR would lead to a better understanding of this simple concept of Schottky junctions. Transparent GR conducting films were satisfactorily included in thin-film CdTe solar cells as the front electrode [232]. A four-layer GR film was reposed by an ambient pressure CVD method, and it possessed a carrier mobility of 550 cm^2^/V-s and an optical transparency of 90.5%. SOCl_2_ and HNO_3_ are used as the p-type dopants in graphitic materials [233] and the holes were impacted at the surface of this carbon structure. The power conversion efficiency for GR-Si cells is lower than that of using nanotube with Si, and a value of only 10% is achieved. Shi et al. [234] increased the efficiency of GR-Si solar cell by 14.5% under standard illumination with the provision of colloidal antireflection coating. A simple spin-coating process was used for providing antireflection treatment, and there was a 90% increase in incident photon-to-electron conversion efficiency across the board and a substantial increase in short-circuit current density was also observed. Jiao et al. [235] explored the scope of MoS_2_ films by varying their thickness and temperature as an effective interfacial layer in Si/GR solar cells. Experiments showed an increase in PCE from 2.3% to nearly 4.4% with 80 °C annealed MoS_2_ film, while it dropped to nearly 0.6% at 200 °C. Tung et al. [93] synthesized a surfactant-free nanocomposite comprised of chemically converted GR and CNTs, and this method delivered 240 Ω/sq at 86% transmittance. This technology was reported to be inexpensive, massively scalable and overcome the various limitations of indium tin oxide (ITO) with PCE of 0.85%.

The environmental stability of silicon-based and organic bulk heterojunction (BHJ) solar cells for longer duration is of paramount importance [236]. The solar cell components are continuously vulnerable to air and other chemicals present in atmosphere, which causes change in power conversion efficiency of organic materials as well as in physical properties, namely, carrier mobility, resistance and optical transparency. Currently, GR-based solar cells exhibit PCE efficiencies from 10% to 15%, which is dependent on the selection of materials and solar cell configuration. Xie et al. [237] observed a power conversion efficiency of 10.56% for a five-layered GR/P3HT/CH_3_-Si organic solar cell (OSCs) having a device area of 4 sq. mm. Li et al. [238] experimentally noted a power conversion efficiency of 15.5% with an Al_2_O_3_-coated n-GaAs/GR solar cell and theoretically estimated a PCE of 25.8% for n-GaAs/GR hybrid solar cells. It has been reported that the electrode of polymer solar cells (PSCs) should be of high conductivity and thickness. So, the dominant electrode material applied in PSCs was ITO, a doped n-type semiconductor consisting of ∼10% SnO_2_ and nearly 90% In_2_O_3_ [239,240]. Now ITO is available on the market with 80% transmittance and a low film resistance of 60–300 Ω/sq on polyethylene terephthalate (PET) and 10 Ω/sq on glass. Yin et al. [241] transferred r-GRO onto PET which was further used as conductive and transparent electrodes for flexible organic photovoltaic (OPV) devices. It was observed that when the optical transmittance remains higher than 65%, the output of the OPV devices relies predominantly on the charge transport efficiency through r-GRO electrodes. However, ITO had severe drawbacks: high cost due to the scarcity of indium, difficulties in patterning, complex processing requirements and problem of sensitivity to both basic and acidic environments. Yusoff et al. [242], instead of using ITO, experimented with Au-doped single-layered GR nanoribbons as an electrode in tandem solar structure and achieved a PCE of 8.48%, which was the maximum efficiency for ITO-free tandem OSCs. Zhang et al. [243] demonstrated a TiO_2_-Al composite for modifying mono-layered GR as an effective cathode for OSCs. The results indicated that the modified GR cathode-based composite had increased the PCE from 1.27% to 2.58%.

### 6.2. Perovskite Solar Cells

Perovskite solar cells (PSCs) have been progressing as the latest photovoltaic technology in the current global energy scenario, and researchers are striving to develop stable and efficient perovskite-based devices [244]. Although a lot of progress has been made toward enhancing PCE, the stability of PSCs is still a concern (Figure 8) [245]. The thermal stability and e-poor air (H_2_O and O_2_) adsorption are a major hindrance to the commercial exploitation of PSCs [246,247]. The chemical stability of PSCs is influenced by exposure to water and oxygen, and there is degradation of the perovskite layer, and its hydrolysis takes place, which causes a change in colour from dark brown to yellow. Hence, a blend of perovskite and GR can be considered for increasing the stability and performance of solar cell devices. Girtan and Rusu [248] found that one explanation for the decline in BHJ OSCs is that the interface between the hole extracting interfacial layer poly-3,4-ethylenedioxy-thiophene:poly(styrene sulfonate) (PEDOT:PSS) and the ITO anode causes the failure of solar cells. PEDOT:PSS mixture has been utilized as a hole transport layer (HTL) in organic solar cells, but due to its excessively acidic nature, it exhibits higher chemical reactivity due to the presence of H_2_O molecules in the atmosphere, which causes corrosion of the ITO electrode and reduces the efficiency polymer solar cells [249,250]. Wang et al. [251] disclosed the effect of a TiO_2_/GR layer in PSCs and a maximum PCE of 15.6% was achieved. Feng et al. [252] deposited a monolayer GR film on Cu foil by CVD process, and a Schottky junction solar cell has been made by transferring the prepared layer onto a silicon-pillar-array (SPA) substrate. Such GR/SPA solar cells obtained maximum energy conversion efficiency of 2.90% with a junction area of 0.09 cm^2^. Park et al. [253] fabricated GR electrodes based OPV devices for observing the effect of HTL, GR morphology and counter electrodes. It has been found that the morphology of the HTL wettability and the GR electrode on the GR surface play an imperative role in the effective amalgamation of GR films onto OPV devices. Tong et al. [254] demonstrated the application of CVD grown MoO_3_-modified GR intermediate layer in both parallel and series tandem solar cells. It has been concluded that the PCE of the solar cell may be improved by work function modification of the GR by coating it with metal oxide. Tung et al. [255] found that synergistic GRO/PEDOT:PSS aqueous dispersions results in greatly increased solution viscosity, which can yield an adhesive composite with a significantly higher electrical conductivity. The sticky GRO/PEDOT interconnect layer greatly enables the construction of solution-processed tandem solar cells through direct adhesive lamination, which aids in the eradication of the constraint imposed by orthogonal solubility during solution processing. Wang et al. [256] established an interface engineering technique for deploying GR film as the positively charged electrode in poly(3-hexylthiophene-2,5-diyl):[6,6]-phenyl C61 butyric acid methyl ester (P3HT:PCBM)-based polymer solar cells. With a variation in interface amid the photoactive layer (with PEDOT: PSS and MoO_3_) and GR anode, the cell power conversion efficiency attains nearly 83.3% that of control devices that use an ITO anode. Further, Li et al. [257] used GRO films as the electron-blocking layer and HTL in OPVs, and significant enhancement in OPV efficiency was observed which was almost the same as that which was achieved with devices fabricated with PEDOT:PSS as the HTL. Liu et al. [258] utilized thermally reduced GRO as a hole transport layer for manufacturing BHJ solar cell devices, and it was found that the conjugated structure of GR is influenced by annealing temperature, and it affects the electrical conductivity of GRO. Solar cell devices having 130 °C reduced GRO, as the HTL showed lower fill factor than devices having 230 °C r-GRO as the layer. Yin et al. [259] deposited monocrystalline ZnO nanorods on highly conductive r-GRO films on quartz. Feasibility study showed that the obtained ZnO nanorods on r-GRO were used for fabricating organic–inorganic hybrid solar cells with a layered configuration of ZnO nanorods/r-GRO/quartz/P3HT/PEDOT:PSS/Au. The observed PCE ≈ 0.31% was noticed to be more than that which was observed with earlier solar cells in which GR films were used as electrodes.

### 6.3. Properties Based Applications

Despite GRO not being a good electrical conductor, the conjugated network can be put to reduction in hydrazine vapour or by high heat after deposition [260]. However, both of the reduction approaches exhibit their own demerits, as flexible substrates such as PET are not suitable at higher temperatures, and hydrazine vapors are capable of accessing and reducing only the outer surface of deposited films. Efforts were made to blend chemically converted GR and CNTs into a single layer, but the resultant film thickness was higher and not suitable for optical applications. To overcome this issue, Tung et al. [93] dispersed CNTs in anhydrous hydrazine for the first time and yielded the manufacturing of a nano composite consisting of CNTs and chemically reduced GR which delivered a 240 Ω/sq film resistance at 86% transmittance. They demonstrated a PCE of 0.85% in the feasibility study of a polymer solar cell. Wang et al. [261] employed GR-based films which were obtained by thermal reaction of synthetic nano GR molecules of giant polycyclic aromatic hydrocarbons (PAHs) as a window electrode in OSCs. The developed GR film showed increased interactions with the substrate in comparison to GRO-induced film. The 4 nm-thick film had 90% transparency at a wavelength of 500 nm. Apart from the feature of acting as a transparent conductive electrode, GR is also useful in PV devices due to other promising features. GR has been used in conjugated polymers for improving the charge transportation and exciton dissociation features of the materials [262]. Yong et al. [263] theoretically computed the expected efficiency of nanosized GR-based PV devices and found an efficiency of more than 12% for single-cell and approximately 24% in case of tandem structure.

Solar cells are economical, stable and efficient energy sources, which are of paramount importance during this period of energy crisis. The dye-sensitized solar cell (DSSC) is considered as the most encouraging substitute for conventional inorganic semiconductor photovoltaic devices. Yan et al. [264] deployed GR quantum dots (QDs) as a solute sensitizer for DSSCs. SCC and OCV of 200 μA/cm^2^ and 0.48 V, respectively, were reported with a fill factor of 0.58. These values of fill factor and open-circuit voltage were approximately the same as those found in cells sensitized by ruthenium complexes. Guo et al. [265] used a chemically reduced thin GR layer which was electrodeposited on ITO-coated glass for immobilizing CdS QDs for QDs sensitized solar cells. It has been demonstrated that GR/CdS QDSC exhibits an incident PCE of 5%, which is much superior to that of single-walled CNTs/CdS (0.45%)-based devices. Further, it has been reported that the utilization of chemically reduced GR is helpful in promoting a homogeneous dispersion of CdS QDs on the electrode for decreased contact resistance and higher absorption of light. Wang et al. [266] demonstrated ultrathin, conducive and transparent GR films as a substitute for the universally deployed metallic oxides electrodes for DSSCs. Such GR films exhibited transparency and electrical conductivity of ≥ 70% and 550 S/cm, respectively, for sizes of 1000–3000 nm. Li et al. [267] developed vertically aligned CNTs (VACNTs) directly on a free-standing GR paper. Electrodes made of this material delivered excellent performance in LIB and DSSCs. This may be due to the excellent conductivity and mechanical properties of GR paper, the beneficial carrier transportation capability of VACNTs and the firm bonding between the free-standing paper and nanotubes. Yang et al. [268] incorporated the chemically reduced GR into a TiO_2_ nanostructure photo anode as a two-dimensional bridge for DSSCs. Such DSSCs yielded short-circuit current and incident PCE improvement by 45% and 56%, respectively, achieving an increment of PCE from 5.01% to 6.97%. Roy-Mayhew et al. [269] proposed a novel electrochemical impedance spectroscopy equivalent circuit which confirms the experiential spectra characteristics to the suitable phenomenon to understand the catalytic action of fGR sheets toward the reduction of triiodide. It was demonstrated that such f-GR sheet-based ink can be used as an excellent electrode material. Kavan et al. [270] accumulated GR nanoplatelets by the way of thin semi-transparent film on F-doped tin oxide (FTO) which showed higher electro catalytic activity for Co(6-(H-pyrazol-1-yl)-2,2′-bipyridine)_2_. DSSCs with Y123 dye adsorbed on titanium oxide photo anode delivered PCE in the range of 8–10% for both nanoplatelets as well as platinum-based electrodes. However, better performance was achieved with a GR nanoplatelets cathode than that with a Pt-FTO cathode. Song et al. [271] demonstrated a novel and facile technique to improve the fill factor and photocurrent, and thus the overall PCE of an organic DSSC by the incorporation of an r-GRO layer between the dye and TiO_2_, which was attributed to the development of a TiO_2_-r-GRO Schottky junction in the proposed DSSC device.

GR and its related products are environmentally friendly and exhibit high charge mobilities which can be utilized for the collection of charge in solar cells. However, such useful characteristics have not been fully utilized in PV applications because of very low solubility and inclination to aggregating into graphite. Yan et al. [264] demonstrated a new solubilisation strategy for large GR nanostructures. They synthesized uniformly sized, solution-processable, black GR-QDs by solution chemistry, and they have been proved to be potential candidates for DSSCs. It is likely that other graphene-based OSCs offer tremendous opportunities and potential for improving photovoltaic performance. Besides increasing PCE, other important challenges to consider are cost-effectiveness, environmental stability against chemicals and photo-oxidation, and eco-friendly, nontoxic GR-based solar cells, which should be examined for large-scale commercial production and applications. Previous photovoltaic devices which utilize GR or GR-based materials have been compiled in Table 6.

## 7. Nanolubricants

Wear and friction are the primary reasons for energy loss in moving components. It has been reported that 33% of fuel energy is wasted because of frictional losses in different components such as brakes, tires and transmission, etc., [13,272]. Lots of research and development is going on for improving the characteristics of lubricants, but still, billions of dollars are lost due to friction in the automobile sector and other industries. As seen in Figure 9, since 1995, there has been a significant diversification of this field of study across numerous disciplines [13].

### 7.1. Design and Development

The need of the hour is to develop efficient lubricants which can effectively reduce energy losses. A recent trend is to use nanoparticles in lubricants for performance enhancement [273]. It has been reported that the inclusion of nanoparticles in lubricants, even at very small concentrations, substantially reduced the wear and friction coefficients. Different types of nano-lubricant mechanisms have been shown in Figure 10 [274]. The unique properties of economically friendly GR and CNTs have considerably attracted the attention of researchers to study their effects as nano-additives to lubricants for improving tribological characteristics. The blending of nanoparticles into different oils is a problematic task, as their shape, characteristics, size, concentration and chemistry with base lubricant have to be studied in detail. It has been shown that nanoparticles tend to get coagulated or agglomerated because of their high surface energy in most of liquids, specifically when there are temperature or pressure variations [275]. Despite having excellent electrical, optical and thermal characteristics, GR can also act as a lubricant in solid or colloidal form [276]. Numerous analytical and experimental investigations performed at both the microscale and nanoscale levels have revealed that GR can be used in solid form or as a nano-additive for the substantial reduction of friction [277,278]. It has been observed that the use of two-dimensional materials such as GR as a friction modifier is dependent on a number of parameters such as morphology, thickness, manufacturing technique and surface chemistry. GR has excellent tribological properties due to its chemically inert nature, high extreme strength, and capability of easy shearing on its compactly packed and atomically smooth surface. In 2009, Filleter et al. [279] used GR as a solid lubricant and established that wear and friction coefficients were decreased due to a reduction in the thickness of commonly used solid lubricants, graphite by many layers [280]. Guo and Zhang [281] added multi-layered GR in polyalphaolefin-2 (PAO2) in various proportions for estimating the COF with nano-lubricants for a steel–steel contact on a four ball tribotester. COF was found to be decreased for all the nano-lubricants irrespective of GR concentration in comparison to PAO2, and the best results were achieved with the smallest concentration (0.05 wt%) of dispersed GR. Senatore et al. [282] analysed the performance of GRO-based mineral oil and observed that the friction coefficient decreased by 30% because of tribofilm formation. The addition of GR nano platelets on a calcium lubricant has also been reported to reduce the friction coefficient [283]. Elomaa et al. [284] dispersed GRO in water and found that the COF decreased by 57% in comparison to pure water on 1 wt% addition of GRO and upon application of a 10 N load. Lin et al. [285] observed that, with the addition of just 0.075 wt% of GR nano platelets, oil characteristics were enhanced, but such ultra-fine additives cause problems of aggregation. Due to this behavior, the movement and entering of additives in matching surfaces is restricted, which leads to unstable tribological characteristics. The characterization results exhibited that the performance enhancement can be achieved because of the extremely thin laminated structure of GR, which easily makes entry between contact surfaces. In another study, GR was dispersed in synthetic oil for making a stable nano-lubricant, which was utilized for the reduction of wear in a bronze material having textured dimples on its surface [286]. Four plates having textured areas in proportions of 0, 5, 10, and 20%, were tested for studying tribological characteristics. It has been inferred that the nano-lubricant has an enormous potential to enhance the wear-resistant properties of the plates with textured surfaces. COF was decreased by 78%, while the wear rate was decreased by 90% with the use of nano-lubricant. Marchetto et al. [287] carried out the comparative tribological analysis of steel–bronze and steel–iron with a ball on disc tribotester by using blended GR flakes in ethanol suspension. It was observed that GR flakes get bound to native iron in bronze–steel and form a layer in iron–steel interface. As a steel ball with small a contact area develops a tribofilm, friction coefficient is reduced by 48% with the addition of GR flakes.

### 7.2. Hybridization

It has been observed that the dispersion of particles gets improved with the addition of molecular ligands, but they have the disadvantage of getting degraded under high speeds and loads. It is always beneficial to use additives in which no surfactant is needed. For this, GRO helps to ensure uniform dispersion without agglomerating in base lubricant. However, there are certain functional groups (normally, -OH, epoxy and -COOH) which improve particle dispersion in nonpolar solvents. Mungse and Khatri [289] blended GRO nanoparticles in 10 W-40 oil and studied the tribological properties for steel–steel surfaces and found reduction in wear and friction coefficients by 37.5 and 36.4%, respectively. Kiu et al. [290] developed a vegetable oil-based nano-lubricant through the addition of GR nanosheets in concentrations of 25, 50 and 100 ppm. Experimentation with a four-ball tester showed the reduction in friction coefficient at 25 and 50 ppm, while COF increased at 100 ppm concentration in comparison to base oil. Optimum results were achieved with a concentration of 50 ppm. Eswaraiah et al. [291] dispersed 0.025 mg/mL of GR in engine oil and reported 33, 40 and 80% improvement to anti-wear, extreme pressure and frictional characteristics. These improvements were explained by the ball bearing effect of GR particles between moving parts. Wu et al. [292] examined the tribological characteristics of GRO nanoparticles blended with oil-in-water emulsion through experimentation using a ball-on-ring tribotester and found that wear and friction coefficients were reduced by 21.8 and 27.9%, respectively. Cho et al. [293] noticed that the morphology of GR particles plays a crucial role in the strength of GR adhesion on friction surfaces. The results obtained by the use of crumpled GR balls in a poly-α-olefin (PAO) oil showed that wear and friction characteristics were improved with the presence of GR balls [294]. Further, Cai et al. [295] mixed GR nanoparticles in PAO4 oil and monitored the performance under the parameters of textured/untextured surface for pure PAO4/GR-blended PAO4 lubricants. Materials having a 10% textured area along with GR-based lubricant performed with the best anti-wear characteristics. Azman et al. [296] examined the influences of GR nanoplatelets as additives in palm-oil trimethylolpropane (TMP) ester blended in PAO. At 0.05 wt% concentration of nanoplatelets, COF and wear rate were observed to be reduced by 5% and 15%, respectively,

### 7.3. Characterization Based Studies

Cheng and Qin [297] demonstrated that GR-based nano-lubricant brought reduction in friction coefficient in the range of 40–60%, while the wear coefficient was also decreased by more than 50%. Further, Kumar and Wani [298] found that GRO in ethanol and SAE20W-50 oil reduced the wear rate by 60–80%, and moreover GRO was recommended as an effective lubricant nanoadditive due to its eco-friendly nature. Vidal and Avila [299] examined the influence of blending nano graphite platelets in mineral-based oil, which revealed the substantial increase in anti-friction and wear characteristics. Liang et al. [300] observed the friction and wear characteristics of exfoliated GR-blended aqueous lubricants which were prepared by the addition of Triton X-100 as a surfactant. Such aqueous solutions outperformed GRO at the same concentration and reported a reduction in COF and wear scar diameter by 81.3 and 61.8%. Zhang et al. [301] used oleic acid for surface modification with liquid-phase exfoliated GR, which was added by 0.02 to 0.06 wt% concentration of lubricant. There was a reduction in wear scar diameter and COF by 14% and 17%, respectively. Meng et al. [302] observed that there was a reduction in wear and COF by 52.7% and 27%, respectively, at 0.05 wt% of the Cu nanoparticle-decorated GRO composite in liquid paraffins. It was suggested that Cu nanoparticles, due to their small anchoring on the GRO nanosheets, improved lubrication characteristics. In another study, they dispersed Ag nanoparticle-blended GR nanocomposite in multi-grade (10W40) engine oil [303]. It was found that the interlamination gaps of GR sheets were increased due to the presence of silver nanoparticles, and further, they prevented restacking of the GR sheets during rubbing, which enhanced lubrication performance. Gupta et al. [304] produced r-GRO nanosheets through the oxidation of graphite succeeded by hydrazine treatment for the reduction of the oxygen functionalities. The obtained nanosheets were functionalized with poly (ethylene glycol) 200 (PEG200). When 0.03 mg/mL functionalized nanosheets were dispersed in PEG200, the COF and wear rate were reduced by 38% and 55%, respectively. It was also found that, at high r-GRO concentrations, the lubrication efficiency was reduced as a consequence of GR–GR inter-sheet collisions, producing chemical defects and mechanical energy at contact interfaces. Chen et al. [305] investigated the tribological characteristics for few-layered GRO nanosheets, which were mixed in oil, by using a ball-on-disc tribotester at 373 K. There was a 10 to 20% decrease in COF at different loading conditions with a 0.5 wt% concentration of GRO in base oil. Kinoshita et al. [306] found that with the addition of GRO nanoparticles in water-based coolant, the friction coefficient was decreased to 0.05 and the surface wear was almost eliminated with running of 60,000 cycles. Zhou et al. [307] demonstrated the friction variation at the different concentrations of r-GRO/ZrO_2_ nanocomposite in paraffin oil with time and found that there was a reduction in COF with nanolubricants. However, it was also observed that when the concentration was increased by 0.2 wt%, the friction coefficient also increased. Ou et al. [308] demonstrated that r-GRO exhibits good anti-wear and anti-friction ability because of the intrinsic structure of GR and its self-lubricating property. Fan and Wang [309] and Wu et al. [292] dispersed modified GRO in multi-alkylated-cyclo-pentanes and water–oil emulsion, and the results exhibited a reduction in COF by 27% and 18%, respectively. Wrinkled paper-like GRO sheets and fluffy structured functionally r-GRO nano-sheets have been prepared to achieve oil stability [310]. Similarly, the characterization results revealed that for functionalized r-GRO, there was a wide hump in place of sharp peaks at 19.9° caused by functionalization. Wu et al. [311] synthesized and dispersed nanosized MoS_2_-decorated GR nanocomposite in perfluoropolyether base oil. The results obtained by a ball-on-disc tribotester at ambient temperature with vacuum conditions exhibited a maximum reduction of 57.1 and 97% in COF and wear rate, respectively, at 1 wt% concentration. The researchers have suggested that the MoS_2_–GR nanocomposite yielded significantly enhanced lubrication results as compared to GR, MoS_2_ or a mixture of MoS_2_ and GR because of surface film formation. Lee et al. [312] found that there is an increase in friction with the reduction of GR thickness, and such behaviour was comparable with other nanomaterials such as HBN, NbSe_2_ and MoS_2_. Kumar et al. [313] synthesised polyacrylamide-grafted f-GRO-based nanocomposites through microwave-based surface-initiated redox polymerization of acrylamide using f-GRO and Ce^4+^ as a redox couple in aqueous medium. The results showed a substantial reduction of the COF by 46–55% and improvement in wear rate by 13–37%, thus qualifying this nanocomposite as an aqueous lubricating additive for tribological application. Cheng et al. [314] synthesized zinc borate-decorated GRO nanocomposite using a liquid-phase based ultrasonic-assisted stripping method and blended in 500 SN oil. The results demonstrated a maximum reduction of 48.2 and 40% in COF and wear scar diameter upon the addition of nanocomposite by 2.0 wt% concentration in lubricant. Anti-friction and wear characteristics were manifested because of tribofilm formation at the surface. Zhang et al. [315] investigated the tribological characteristics of GR and multi-walled CNTs in different lubricating conditions and in a vacuum atmosphere. The presence of GR and CNTs was found to generate excellent lubricating conditions due to the development of nano-level tribological mechanisms. Miura and Ishikawa [316] used a grease comprised of alternately arranged C_60_ monolayers and single GR sheets, and they reported that the included C_60_ molecules were capable of rotating between GR sheets by using 13C-nuclear magnetic resonance measurements. It has also been claimed that this innovative combination was capable of yielding the best results in comparison to all other nano-lubricants explored so far. Although thermal reduction of GR is an excellent method, such GR always has some defects, including obvious folding and wrinkling. Choudhary et al. [317] synthesized alkylated GRs on a mass scale and then dispersed in organic solvents. The lubrication characteristics of hexadecane containing Octadecylamine f-GRO were investigated for tribological properties. The outcomes indicated a reduction in the wear and friction by 9% and 26%, respectively, when 0.06 mg/mL of octadecylamine f-GRO was added in hexadecane. Zhao et al. [318] fabricated a novel synthesis method of thermally reduced graphite oxide in H_2_SO_4_, which showed a lamellar structure without obvious folding and wrinkling. It also exhibited outstanding tribological properties, and the wear rate and COF were found to be decreased by 75% and 30%, respectively, even at a high load (1.86 GPa). Song et al. [319] prepared the in-situ growth of Cu nanoparticles decorated on polydopamine (PDA) f-GRO nanosheets. It was found that soybean oil with 0.1 wt% Cu-PDA-f-GRO nanocomposites exhibited the lowest COF under all of the sliding conditions. However, the characterization results showed that these composites have higher anti-wear capabilities than Cu nanoparticles, GRO and Cu-GRO. Singh et al. [320] dispersed SiO_2_/GRO composite powders in (CH_2_OH)_2_ keeping with the additive’s concentration constant (0.125 wt%). A composite with 10 wt% of GRO in a SiO_2_ matrix showed reduction of COF in (CH_2_OH)_2_ by 38% and reduction in wear by 31% as compared to pure (CH_2_OH)_2_.

### 7.4. Advancements

As the youngest graphene derivative, fluorinated GR (FGR) has drawn immense research interest because of its superb performance [321]. FGR inherits excellent mechanical properties of GR, even though fluorination disrupts the van der Waals forces between the FGR sheets; therefore, FGR can be utilized to enhance the mechanical properties of polymers. High hydrophobicity restricts the use of FGR in aqueous environments despite having excellent characteristics. Ye et al. [322] attempted to overcome this by preparing hydrophilic urea-modified FGR, and characterization results revealed that the urea molecules can covalently functionalize the FGR. With 1 mg/mL concentration of urea-modified FGR aqueous dispersion, the wear rate was reduced by 64.4% as contrasted against pure water. Overall, it has been agreed unanimously that GR plays an excellent role in improving anti-wear and anti-friction characteristics. GR and its derivatives can be added to base lubricants in small concentrations, improving tribological characteristics. However, certain factors such as the morphology, concentration and dispersion capability of nanoparticles in addition to the characteristics of base lubricant need to be studied in detail for use as a prospective lubricant [323]. Carbon nanomaterials as lubricant additives are capable of improving the tribological characteristics of base lubricants, but the viscosity of base lubricants is affected by their inclusion in varying concentrations. This aspect can be explored by reducing the concentration of other additives in plastic lubricants, which is beneficial from an environmental perspective [324]. The effect of GR-based materials on the tribological performance has been summarized in Table 7.

## 8. Automotive Sector

Industries and transport sector across the globe have been assigned the task of reducing CO_2_ and other greenhouse emissions to halt the problems arising out of global warming and so that it is safe to live on the planet. Concerted efforts are being made to reduce carbon footprints and energy losses by moving to clean and sustainable energy sources [272,325]. Internal combustion engines are the heart of the automobile sector and are also used in various machinery and equipment. Various types of frictional losses lower the efficiency and raise fuel consumption. Figure 11 shows the new Ford Mach II’s 45% weight-saving multi-material lightweight vehicle (MMLV) body design in white (BIW) [326].

### 8.1. Applications

Traditional additives have been used in fuels and oils for many years, but in the last decade, nanoparticles have been explored for use as potential additives to improve anti-wear and anti-friction characteristics [327]. In automobiles, major frictional losses occur in IC engines, transmission systems, tires and gearboxes. Lubricants with various carbon-based allotropes have found applications in gearboxes, turbines, IC engines and other mechanical and hydraulic systems. These engine oil-based applications constitute 50% of the total global market [328]. Sarafraz and Hormozi [329] dispersed nanofluids consisting of MWCNTs inside a Chevron plate heat exchanger and investigated the pressure drop and heat transfer characteristics. A higher friction factor and pressure drop were observed with MWCNT/water nano-fluids as compared to the base fluid, but excellent thermal performance was the plus point of using nanofluids. Srinivas et al. [330] analysed the thermophysical and heat transfer characteristics of nanofluids using MWCNTs in concentrations of 0.025, 0.05 and 0.1% with deionized water and sebacic acid. An air-cooled heat exchanger, which worked similarly to an automotive radiator, was used for studying the heat transfer performance of nanofluids. There was an 87.3% increase in the overall heat transfer coefficient in comparison to base fluid. M’hamed et al. [331] found that the MWCNT- C_2_H_6_O_2_/H_2_O nanofluids increased the average heat transfer by 196.3% for 0.5 vol% nanoparticle concentration in comparison to base coolant in an automotive radiator system. The findings also revealed that the thermal conductivity, aspect ratio and specific surface area were increased, and that the thermal resistance was reduced with MWCNT-based nanofluid in comparison to base fluid. In contrast to other carbon-based materials, GR-based nanomaterials seem to have excellent potential for use as lubricant additives due to their excellent properties, and it has become a remarkable research subject.

### 8.2. Innovations

Researchers have been continuously exploring the possibility of using GR as a nanoparticle additive in automobile lubricants. Rasheed et al. [327] explored the performance of GR-based engine oil nano-lubricant on a test rig equipped with a four-stroke engine. The length and flake size of GR were found to have an effect on the thermal performance of nano-lubricants. The researchers observed that a few-layered GR is instrumental in reducing COF and improving engine performance. Ramón-Raygoza et al. [332] used multi-layer graphene (MLG) for preparing nano-lubricants such as MLG-Cu and MLG- PANI to be used in automobile engines. It has been observed that there was a substantial decrease of 63% and 43% in the wear and friction coefficients, respectively, with MLG-Cu dispersions in base oil. Selvam et al. [333] computed the thermo-hydraulic performance of GR nanofluids/H_2_O-C_2_H_6_O_2_ (70:30) with 0.1–0.5%. volumetric concentrations. At 0.5% concentration, there was about a 39% and 104% increase in the pumping power and overall heat transfer coefficient, respectively. Many other experimental studies have been carried out checking the effect of nanofluid on automotive cooling systems, but some conflicting results have been observed. Contreras et al. [334] computed the thermo hydraulic performance of nanofluids, containing Ag-GR nanoparticles with a binary mixture having equal proportions of H_2_O and C_2_H_6_O_2_ as a base fluid in automotive radiators. Silver-based nanofluids were observed to increase the heat transfer rate by 4.4%, but there was a reduction in the thermo-hydraulic performance with GR-based samples in comparison to base fluid. Esquivel-Gaon [335] demonstrated that the use of r-GRO as nano additives contributes to decreasing the environmental influence of the transport sector, which was a positive step toward a more sustainable automobile sector. Izzaty et al. [336] elaborated on the industrial perspectives on the implementation of graphene composites in the automotive industry. They focused on three different aspects of innovation management, namely PESTEL (political, economic, social, technological, environmental and legal) analysis, business ecosystem and scenario planning. It has been inferred that a strategic analysis based on PESTEL factors revealed two major concerns in the automotive industry: the sustainability of supply chains and the quality of GR composites. Kojima et al. [337] demonstrated a GR Hall sensor manufactured by use of conventional Si process technology for automotive applications. It was reported that their sensor meets the demands for high precision and good usability with a sensitivity of 0.1 V/VT, a thermal coefficient of sensitivity of 2800 ppm/K and a mobility of about 2000 cm^2^/Vs. Toh et al. [338] examined the thermal performance of H_2_O-based GR nanoplatelets (GRnP)-based nanofluid at varying volumetric concentrations and temperatures in automotive radiators. The thermodynamic performance of a flat tube radiator containing a GRnP base lubricant was simulated using ANSYS Fluent software. It was inferred that the enhancement in average Nusselt number of 74.18% at 0.5 vol%, could improve the efficiency of automobile cooling systems, and this led to a smaller-sized radiator resulting in the improvement of fuel efficiency for the engine. Sumanth et al. [339] studied the effect of the nanofluid carboxyl GR, which was added to C_2_H_6_O_2_ (50:50 vol%)-distilled water at different concentrations, on the performance of an automotive radiator. It was found that by adding carboxyl GR nanoplatelets, the Nusselt number was increased, and radiator efficiency improved while there was no change in friction factor. The effectiveness of radiator was observed to be increased by 27.38% for 40 °C inlet temperature. Ali et al. [340] used a GR-based nano-lubricant and studied the mechanism of worn surfaces under a boundary lubrication regime and found energy saving in automotive engines due to a self-healing effect of nano-lubricants. The experimental findings indicate that GR nano-lubricant leads to a 17% decrease in the consumed cumulative fuel mass; moreover, the exhaust emissions results showed a reduction of 2.79–5.42% in CO_2_, HC and NOx gasses when the engine was lubricated by GR nano-lubricant. Selvam et al. [341] analyzed the coefficient of convective heat transfer and pressure reduction of GRnP dispersed in mixture of H_2_O-C_2_H_6_O_2_ which flows through an automotive radiator. There was an increase in the coefficient of convective heat transfer of nanofluids on increase in loading of GRnP, mass flow rate and nanofluid. Also, the enhancement of convective heat transfer coefficient for the highest concentration (0.5 vol%), and the highest mass flow rate (100 g/s) was found to be 20% and 51% at nanofluid inlet temperatures of 35 °C and 45 °C, respectively. Amiri et al. [342] developed a high-performance engine coolant by dispersing the GR nanoplatelets in H_2_O-C_2_H_6_O_2_ media. The results revealed the negligible increase in the pressure drop at various temperatures and concentrations, lack of corrosive condition, low friction factor, and a performance index larger than 1. As there was no momentous change in the pumping power in the presence of GRnP-WEG, it was concluded that such a coolant can be used in place of conventional coolant in thermal equipment.

GR-based nano-lubricants have an enormous potential for increasing fuel efficiency due to the reduction in frictional losses and adapting to the requirements of automotive engines. Due to superb performance in boundary lubrication regime, GR nano sheets act as solid lubricants on worn surfaces and increase the lifespan of engine parts and enhance the durability of engines [343]. GR has the characteristic of weak van der Waals forces and electrostatic interaction forces between the two-dimensional layers, which enhances the stability of nano lubricants [344]. Further research efforts are paramount to perceive the exact behavior of GR-based nano lubricants from a heat transfer point of view, so that steps can be taken towards the commercialization of GR-based lubricants in the automotive sector.

## 9. Conclusions

This paper gives an overview of different graphene-based materials in the areas of energy sustainability and environmentally friendly applications.

(a)Graphene, due to its unique characteristics, has been put to multifarious uses by researchers for developing and designing energy saving, conserving and storage devices.(b)Various composite materials have been fabricated by using different derivatives of graphene with PANI, PEDOT and numerous metallic oxides. These materials have been deployed in supercapacitor applications, which resulted in improvement of specific capacitance and power density. Initial research has shown that graphene composite materials can be effectively utilized as electrode materials in Li-ion, Na-ion and Ca-ion batteries for improving the energy density, cyclability and capacity of rechargeable batteries.(c)Researchers have developed a diverse range of graphene-based fuel cell membranes, and their use resulted in the improvement of mechanical properties, ionic conductivity, chemical stability, power density and better water uptake.(d)The development of solar cells by inculcating the fascinating properties of graphene is an area which is rapidly catching the attention of researchers for improving the photovoltaic performance of solar cells.(e)Graphene, in different morphologies, has been blended with various oils/fluids, which has resulted in improvements of tribological characteristics and increased environmental friendliness.(f)Use of graphene-based lubricants has been well established in boundary layer lubrication, which has resulted in the reduction of frictional losses and enhanced product life by decreasing wear rate.(g)The excellent heat transfer characteristics of graphene in different fluids has prompted its use in the automotive sector for the rapid dissipation of heat which is generated during the functioning of engine and transmission units.

This comprehensive survey will serve as a valuable guide for researchers in selecting a specific form/morphology of graphene product and its synergy with various materials for developing composites which can be used in fuel/solar cells, supercapacitor applications and rechargeable batteries. Commercial usage of graphene is possible with economical production of graphene in different forms. Experimental investigations of graphene composite electrodes and membranes need to be carried out in order to check the long-term effects on the performance of batteries and fuel cells. There is a need to explore use of graphene-based nano lubricants in hydrodynamic lubrication regime. When these key points are answered, graphene can be effectively utilized as a potential energy-saving material.

## Figures and Tables

**Figure 1 materials-15-06241-f001:**
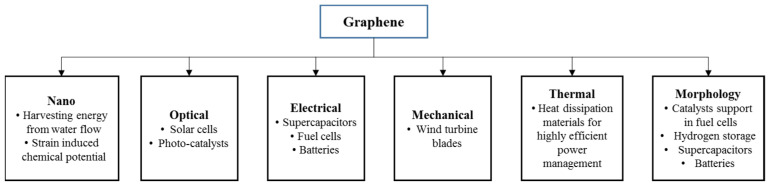
Correlation between graphene characteristics and their applications in energy solutions.

**Figure 2 materials-15-06241-f002:**
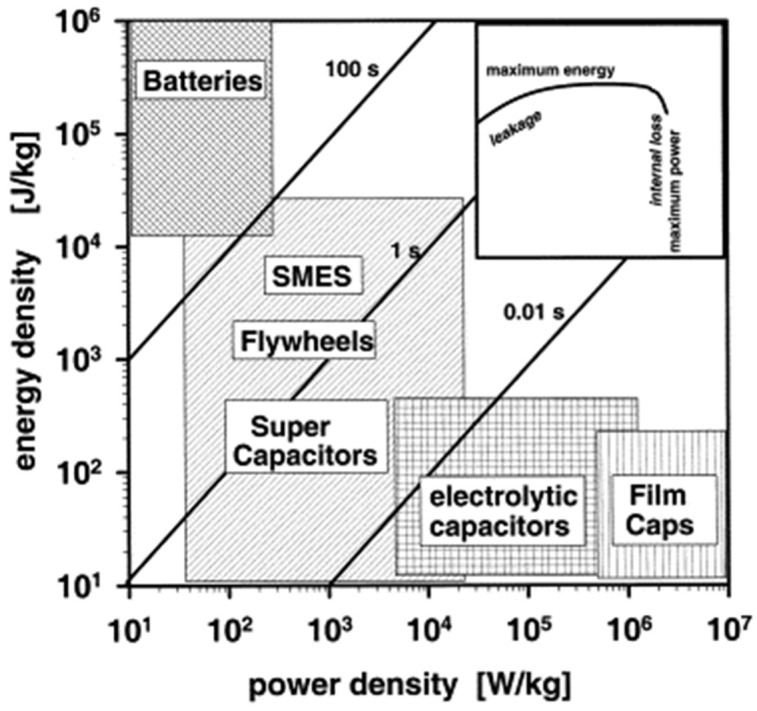
Ragone scheme showing energy loss owing to internal dissipation and leakage losses for sufficiently high and low power (reproduced from [46] with permission from Elsevier).

**Figure 3 materials-15-06241-f003:**
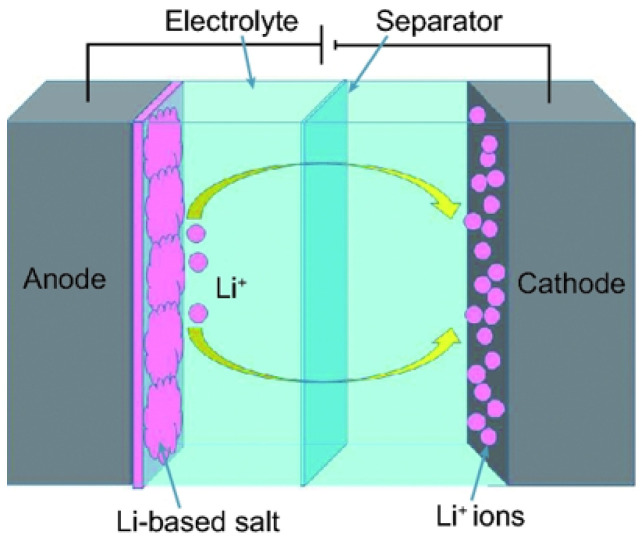
Schematic diagram of a hybrid supercapacitor [92].

**Figure 4 materials-15-06241-f004:**
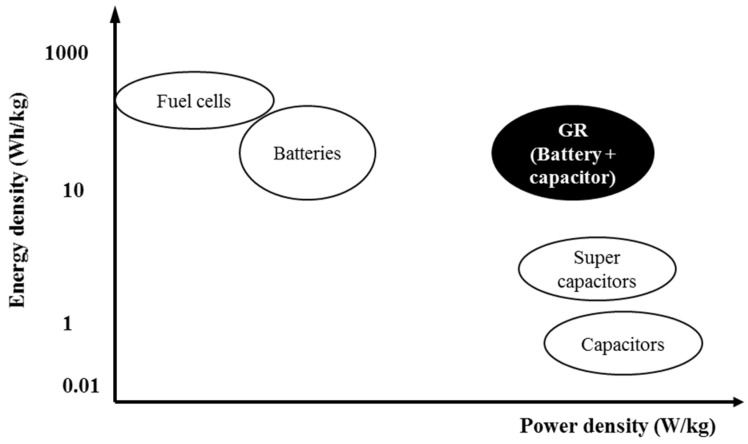
Ragone plot for energy storage devices.

**Figure 5 materials-15-06241-f005:**
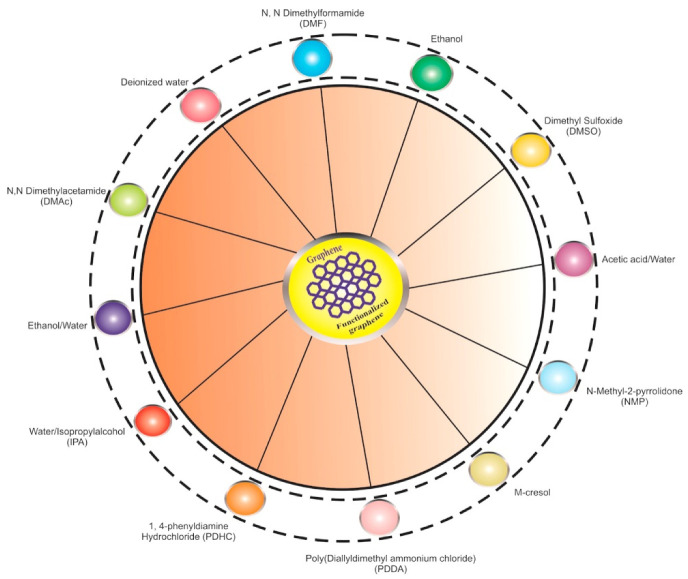
Graphene oxide-compatible solvents for polymer electrolyte membrane fuel cells (reproduced from [167] with permission from Elsevier).

**Figure 6 materials-15-06241-f006:**
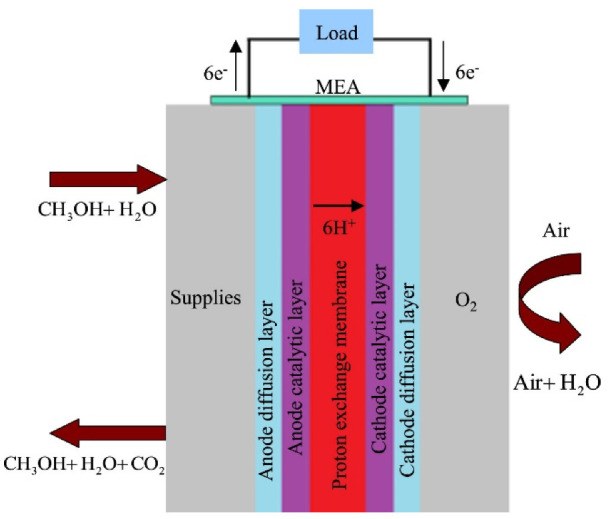
Scheme of the basic anatomy and functioning principle of a DMFC (reproduced from [199] with permission from Elsevier).

**Figure 7 materials-15-06241-f007:**
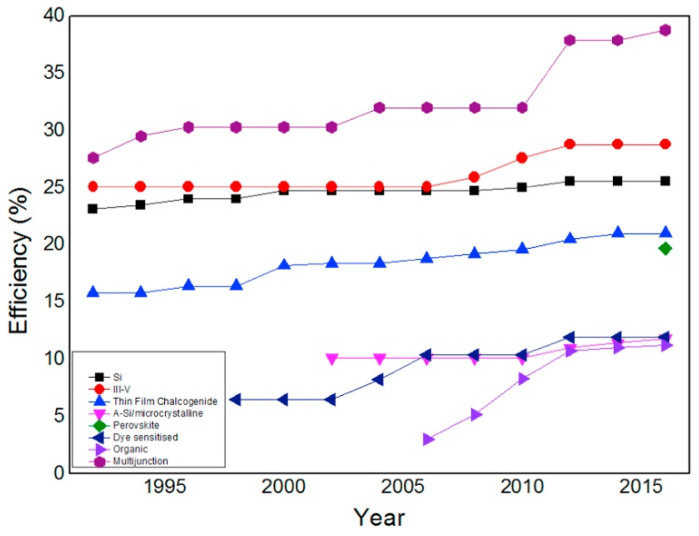
Solar cells’ efficiency reported based on several technologies (reproduced from [224] with permission from the Elsevier).

**Figure 8 materials-15-06241-f008:**
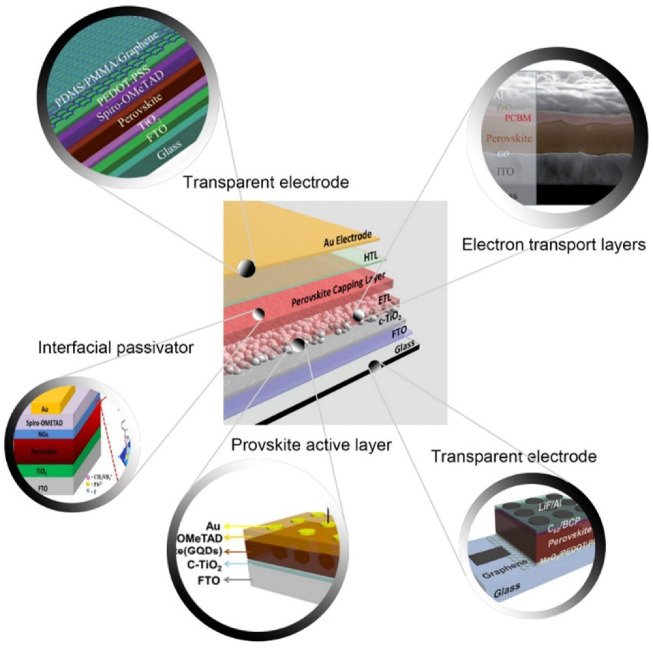
Schematic illustration of graphene and its derivatives in polymer solar cells (reproduced from [245] with permission from Elsevier).

**Figure 9 materials-15-06241-f009:**
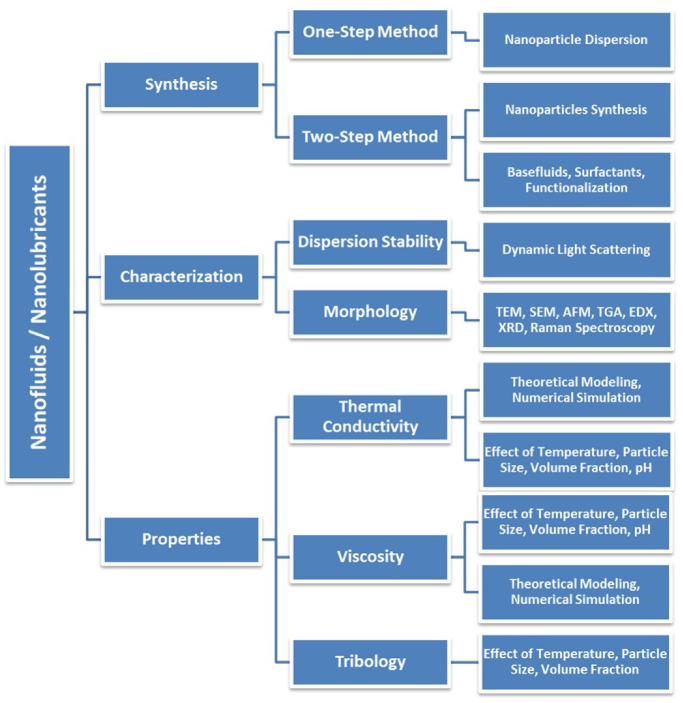
Nanofluids and nano-lubricants research areas and methodology (reproduced from [13] with permission from Elsevier).

**Figure 10 materials-15-06241-f010:**
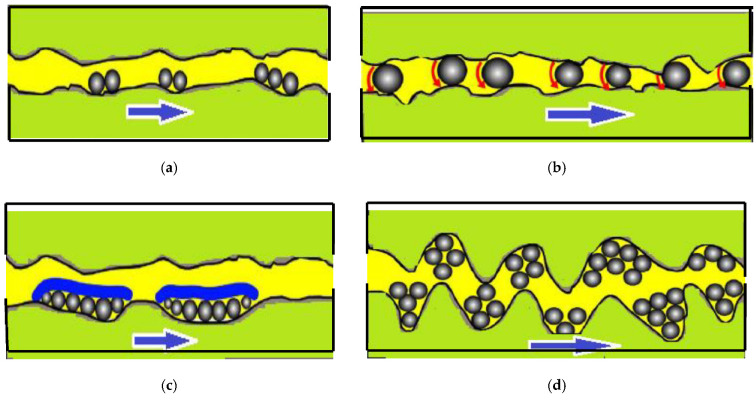
Schematic illustration of different lubrication mechanisms of nanoparticles [288]. (**a**) Polishing mechanism, (**b**) rolling mechanism, (**c**) self-repairing mechanism, (**d**) tribo-film mechanism.

**Figure 11 materials-15-06241-f011:**
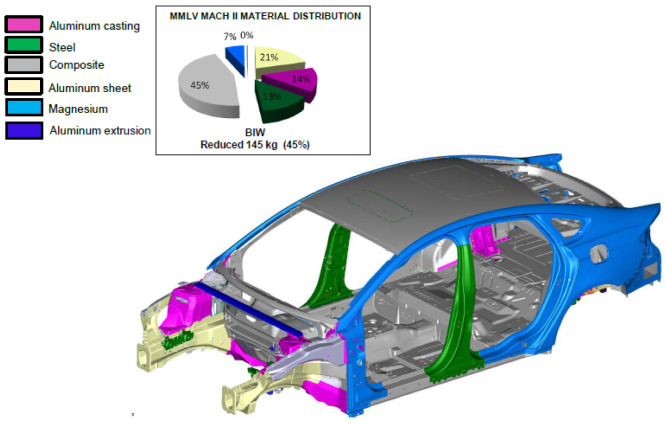
Multimaterial lightweight vehicles MMLV [326].

**Table 1 materials-15-06241-t001:** Properties of single-layered graphene [11,12,13].

Property	Value	Property	Value
Bond type	sp^2^	Crystal structure	Hexagonal
Dimension	2D	Surface area	2630 m^2^/g
Melting point	~3852 °C	C-C bond length	0.142 nm
Mobility (typical)	~200,000 cm^2^V^−1^s^−1^	Mobility (intrinsic)	10^8^ A cm^−2^
Real density	2.25 g/cm^3^	Mass (bulk) density	~0.3 g/cm^3^
Relaxation length	~15,000 cm^2^V^−1^s^−1^	Thickness	~1–2 nm
Thermal conductivity	~5000 W/m-K	Electrical conductivity	~20,000 S/cm
Elasticity modulus	~1 TPa	Intrinsic strength	~130 GPa
Fracture toughness	~4 MPa-m^0.5^	Breaking strength	42 Nm^−1^
Electron mobility	~2.5 × 10^5^ cm^2^/(V·s)	Electron density	2 × 10^11^ cm^−2^
High temp. resistivity	−75 + 200 °C between not changing	Optical transmittance	97.7%
Interplanar spacing b/w Gr sheets	0.335 nm	Spin R	1.5–2 µm
Fermi velocity	300–500 nm	Phase coherence length	3–5 µm
Current density	c/300 = 1,000,000 ms^−1^	Sheet resistance	1.3 × 10^−4^–5.1 × 10^−4^ Ω/sq

**Table 2 materials-15-06241-t002:** Summary of GR based materials for capacitor applications.

Materials	Process/Electrolyte	Power Density (kW kg^−1^)	Energy Density (W h kg^−1^)	Specific Capacitance (Fg^−1^)	Ref.
r-GRO	Reduced GR and convection dry	9.8	85.6	250	[53]
PPy-GR	Electric deposition on GRO	3	5.7	1510	[58]
PANI-GR	In situ polymerization	0.14	37.9	1126	[80]
PEDOT-GR	Oxidative polymerization	0.038	12	304	[59]
MnO_2_-exfolitated graphite	Dip and dry deposition	110	12.5	315	[89]
RuO_2_-GR	Sol-gel treatment with RuO_2_ and GRO	0.05	20.1	570	[70]
Fe_3_O_4_-GR	Crystallization of metal oxide with r-GRO	2.4	85	326	[72]
N-doped GR	N_2_ plasma	800	48	282	[75]
N-doped GR	Et_4_NBF_4_/PC (2 at% N)	1	76.7	138.1	[77]
N-doped GR	KOH (10.1 at% N)	7.98	25	326	[78]
N-doped GR	Bu_4_NBF_4_ (10 wt% N)	-	-	248.4	[79]
PANI-GR	in situ polymerization/reduction–dedoping/redoping	136	34.8	1126	[80]

**Table 3 materials-15-06241-t003:** Application of GR based electrode materials for LIBs.

Material	Type	Capacity (mAh/g)	Cycling Condition	Ref.
3D GR integrated LiFePO_4_	Cathode	146	17 mA/g	[128]
LiFePO_4_/3D GR Composite	Cathode	160	C/3	[129]
LiFePO_4_@C/r-GRO	Cathode	119	20 C	[130]
Free-standing 3D GR/LiFePO_4_	Cathode	115	10 C	[132]
N-doped GR/LiFePO_4_	Cathode	78	100 C	[131]
GR/LiMn_2_O_4_	Cathode	113	0.5 C	[136]
GR/LiMn_2_O_4_ Nanoparticles	Cathode	140	70 mA/g	[138]
3D macroporous GR-based Li_2_FeSiO_4_	Cathode	313	0.1 C	[142]
GR and carbon nanotube co-modified Li_3_V_2_(PO_4_)_3_/C	Cathode	147.5	20 C	[121]
GR@Si@GR 3D sandwich structure	Anode	2515	0.4 C	[116]
3D GR/SnO_2_	Anode	1096	1 A/g	[106]
Li_4_Ti_5_O_12_/N-reduced graphene oxide	Anode	117.8	30 C	[122]
Graphene/SnO_2_-Co_3_O_4_ Nanocubes	Anode	1665	100 mA/g	[112]
MoS_2_/graphene	Anode	870	1 A/g	[117]
3D graphene/SnS_2_	Anode	1386.7	100 mA/g	[118]
3D graphene/SnCo Nanoparticles	Anode	1117	72 mA/g	[119]

**Table 4 materials-15-06241-t004:** Charge transfer for Na/Ca to graphene [148].

Ion	Divacancy	Pristine	Stone-Wales
Na^+^	0.8848 e	0.6617 e	0.8073 e
Ca^2+^	1.3727 e	0.8208 e	1.1189 e

**Table 5 materials-15-06241-t005:** GR based membranes in fuel cells.

Polymers	GR (Content)	Solvent	Contribution	Ref.
Graphene based membranes in PEMFCs				
Poly (ethylene oxide) (PEO)	GRO (0.5 wt%)	Distilled H_2_O	Improvements to the Young’s modulus, electronic resistance tensile strength and ionic conductivity	[172]
Polybenzimidazole (PBI)	Graphite oxide & Sulfonated GRO (2 wt%)	N,N-dimethylacetamide (DMAc)	Enhancement of ionic and proton conductivities	[183]
Sulfonated polyimide	Ionic liquid polymer adapted GR sheets (10 wt%)	Dimethyl sulfoxide (DMSO)	Increase in tensile strength, ionic conductivity and other mechanical properties	[197]
Nafion	Rolled up graphene oxide sheets	N,N-dimethylformamide (DMF)	Enrichment of water retention ability, better proton transport and conductivity, reduced activation energy	[193]
Nafion	f-GRO (5 and 10 wt%)	Ethanol	Improvement to chemical and mechanical stability, better water intake and IEC with higher proton conductivity	[190]
Sulfonated poly(ether sulfone)	Mesonaphth o-bifluorenegraphene moiety	Dimethyl sulfoxide (DMSO)	Enhancement of thermal stability, better water intake and IEC with increased proton conductivity	[194]
Polybenzimidazole (PBI)	3-amino propyl-triethoxysilane ionic liquid f-graphite oxide (5 wt%)	N,N-dimethylacetamide (DMAc)	Better ionic and proton conductivities	[181]
Nafion–SPEEK	GRO (0.75 wt%)	Ethanol/Water(75:25 V/V)	Increase in proton conductivity which results in increase in current and power densities	[191]
SPEEK	Polydopamine-modified GRO (DGRO) (2.5, 5, 7.5, and 10 wt%)	Dimethyl formamide (DMF)	Improvement of the proton conductivity, power and current densities	[203]
Nafion	Graphite oxide (4 wt%)	N,N-dimethylacetamide (DMAc)	Enhancement of proton conductivity and peak power density	[187]
Nafion	Polyoxometalate coupled GRO (1%)	Deionized water	Improvement of the proton conductivity and water retention capacity, decrease in ohmic resistance	[188]
Nafion	GRO Pt-GR (0.5–4.5 wt%)	Water and isopropyl alcohol (IPA)	Increase in tensile strength, but the results obtained with Pt-GR were not optimum	[173]
Nafion	Pt–GR/SiO_2_ (0.5–3 wt%)	Deionized water and IPA	Improvement of cell performance up to 1.5 wt% concentration of Pt-GR, increase in water uptake and proton conductivity	[174]
Nafion	GRO (2, 3 and 5 wt%)	N,N-dimethylacetamide (DMAc)	Increase in tensile strength, water uptake, proton and electrical conductivities	[185]
Nafion/Pt–TiO_2_	GRO (1 wt%)	IPA-water mixture	Enhancement of proton and electrical conductivities	[175]
Graphene based membranes in DMFCs				
Nafion	SGRO (0.5 wt%)	N,N-dimethyl formamide	Improvement of proton conductivity, reduction in activation energy and methanol crossover	[204]
SPEEK	SGRO (0–10 wt%)	N,N-DMAc	Increase in proton conductivity, water retention capacity and mechanical properties, decrease methanol crossover	[205]
SPEEK/PVA	SGRO/Fe_3_O_4_ (3–7 wt%)	N,N-DMAc	Decrease in methanol crossover and improvement in proton conductivity and mechanical stability	[206]
Nafion 115	GRO (0–2 wt%)	Deionized water	Decrease in methanol crossover and improvement in proton conductivity	[210]
Nafion	GRO (0.1–2%)	Dimethyl formamide (DMF)	Retention of ionic conductivity, increase in thermal and mechanical properties, reduction of methanol crossover	[209]
Nafion	PDDA/GRO	-	Improvement of power density and lowering of methanol crossover	[208]
Sulfonated Polyimide (SPI)	Sulfonated propyl-9 Silane GRO	N,N-DMAc	Enhancement of thermal, mechanical, and chemical stabilities, increase in proton conductivity and water retention characteristics	[198]
SPEEK	Carboxyl-functionalized graphene (G(c)) (0.1–0.25 wt%)	DMF	Increase in proton conductivity and reduction in methanol crossover, improvement in water retention capacity and self-humidifying characteristics	[176]
SPEEK	Sodium dodecylbenzene sulfonate (SDBS) adsorbed GRO (5 wt%)	DMF	Improvement of methanol permeability, water retention capacity, proton and electrical conductivities	[202]
Nafion	SGRO (0.05–0.5 wt%)	N,N-DMAc	Reduction in methanol uptake and swelling ratio, increase in proton conductivity and water retention capacity	[189]
SPEEK	GRO (1–6 wt%)	DMF	Improvement of proton conductivity, selectivity and reduction in methanol crossover	[201]
Self-supporting membrane	GRO laminates (3 mg/l)	Deionized water	Better power density in comparison to Nafion membrane without any decrease in open circuit potential	[207]
Others				
SPEEK in Microbial FC	Single layer GRO (0.25 wt%)	N-Methyl−2-pyrrolidone (NMP)	Increase in water retention capability, selectivity, proton conductivity and oxygen diffusion coefficient	[220]
Polybenzimidazole (PBI) in Alkaline Anion Exchange Membrane FCs	Ionic Liquid-GRO (ILGRO)	DMSO	Increase in water uptake, thermal stability, tensile strength, swelling ratio and conductivities	[221]
PVA in Direct Methanol Alkaline FC	GR nanosheets (0.1–1.4 wt%)	Deionized water	Decrease in methanol crossover, improvement in tensile strength and ionic conductivity	[214]
Chloromethylated polysulfone (CMPSU) in alkaline FC	Quaternized graphenes (QGs) (0.25–1 wt%)	DMF	Enhancement of mechanical properties and bicarbonate conductivity	[222]
KOH in alkaline FC	GRO (5 mg/mL)	-	Reduction in hydrogen permeability, improvement in ionic conductivity and peak power density	[223]

**Table 6 materials-15-06241-t006:** Summary of GR and GR-based composites for solar cells.

GR Based Material	Electrode/Function	Sheet Resistance	Transmittance (%)	Category and Configuration of Solar Cells	Power Conversion Efficiency	Ref.
r-GRO	TA	1.8 k Ω/sq	70	Solid-state DSSC: r-GRO/glass/dye/TiO_2_ spiro-OMeTAD/Au	0.26%	[266]
r-GRO	TA	3.2 k Ω/sq	65%	OPV: r-GRO/PET/PEDOT:PSS P3HT: TiO_2_/PCBM/Al	0.78%	[241]
CVD-GR	TA	0.25 k Ω/sq	95%	OPV: GR/quartz/PEDOT:PSS/ CuPc: BCP/C60/Ag	0.85%	[253]
r-GRO-CNT	TA	0.6 k Ω/sq	87%	OPV: r-GRO-CNT/glass/PEDOT:PSS P3HT: Ca:Al/PCBM	0.85%	[93]
CVD-GR	TA	0.08 k Ω/sq	90%	OPV: GR/quartz/MoO_3_^+^ PEDOT:PSS/P3HT:PCBM/Al/LiF	2.5%	[256]
Au-doped GR	TA	0.293 k Ω/sq	90%	OPV: Au-GR/PEDOT:PSS/ P3HT:PCBM/ITO/ZnO	3.04%	[267]
r-GRO	TC	0.42 k Ω/sq	61%	Hybrid solar cell: r-GRO/quartz/ZnO/ P3HT/Au/PEDOT:PSS	0.31%	[259]
CVD-GR	TC	0.22 k Ω/sq	84	Thin film solar cell: GR/glass/CdTe/ CdS/graphite paste/ZnO	4.17%	[232]
Al-TiO2 modified GR	TC	1.2 k Ω/sq	96	OPV: GR/Al-TiO_2_/P3HT:PCBM/MoO_3_/Ag/Au	2.58%	[243]
fr-GRO	CCE	-	-	Liquid DSSC: r-GRO/FTO/dye/I_3_^−^/I_1_^−^/TiO_2_/mediated electrolyte	4.99%	[269]
CNT-r-GRO paper	CCE	-	-	Liquid DSSC: CNT-r-GRO/TiO_2_/FTO/I_3_^−^/I_1_^−^ mediated electrolyte/dye/	6.05%	[267]
GR platelets	CCE	-	-	Liquid DSSC: GR/FTO/Co(III)/(II) mediated electrolyte TiO_2_/dye	9.3%	[270]
GR QDs	Sensitizer of dye	-	-	Liquid DSSC: GR QD/dye/FTO//I3^−^/I1^–^ mediated electrolyte/Pt/TiO2	<0.1%	[264]
GR/n-Si	SJL			Schottky junction solar cell: Ag/GR/Ti/n-Si/Pb/Au	1.65%	[226]
GR-TiO2	SJL			Liquid DSSC: TiO_2_ -GR/FTO/Pt dye/I3^−^/I1^-^ mediated electrolyte	6.06%	[271]
GRO	HTL			OPV: GRO/ITO/Al/P3HT:PCBM	3.5	[257]
MoO3-GR	Interfacial layer			Series tandem solar cell: GR-MoO3/PEDOT:PSS/ITO/P3HT:PCBM/ZnPc:C60/Al/LiF	2.3%	[254]
ZnO-GRO-PEDOT:PSS	Interfacial layer			Series tandem solar cell: ZnO-GRO-PEDOT:PSS/PEDOT:PSS/ITO/P3HT:PCBM/Ca/P3HT:PCBM/Al	4.14%	[255]

(a) TA—transparent anode (b) TC—transparent cathode (c) CCE—Catalytic counter electrode (d) SJL—Schottky junction layer.

**Table 7 materials-15-06241-t007:** Summary of improvements in tribological characteristics using GR based nanolubricants.

Material	Medium	Composition	Reduction in COF (%)	Reduction in Wear (%)	Ref.
GR	Synthetic oil		78	90	[286]
reduced f-GRO	Oil		16	30	[310]
GRO	10W-40 oil		37.5	36.4	[289]
GRO	Oil-in-water emulsion		21.8	27.9	[292]
Exfoliated GR	Oleic acid	0.02 to 0.06 wt%	17	14	[301]
GRO + Zinc borate	500 SN oil	2 wt%	48.2	40	[314]
GR	Grease in semi-solid state	-	40 to 60	50	[297]
Thermally converted graphite oxide	H_2_SO_4_		30	75	[318]
GR	Deionized H_2_O	23.8 to 110 μg/ml	81.3	61.8	[300]
GRO	H_2_O	0.2 wt%	57		[284]
SiO_2_/GRO composite	C_2_H_6_O_2_	0.125 wt%	38	31	[320]
GRO nanosheets	SN150	0.1 wt%	30	-	[282]
ZrO_2_ nanoparticles/r-GRO nanosheets	Paraffin oil	0.06 wt%	56	6.4	[307]
Modified GO	Multi alkylated cyclo pentanes		27	74	[309]
GR nanosheets	Grease	3 wt%	61	45	[283]
MoS_2_/GR nanocomposite	PFPE	1 wt%	57.1	97	[311]
Multi-layered GR	PAO2	0.05 wt%	78	16	[281]
Ag/GR nanocomposites	10W40 oil	0.06 to 0.10 wt%	30.4	27.4	[302]
GR nanosheet	Vegetable based oil	50 ppm	13.5	9.7	[290]
Nanographite platelets	Mineral oil	0.25	17	24.1	[299]
GR	PAO4	0.04 wt%	78	90	[295]
Modified GR platelets	350 SN oil	0.075 wt%	37		[285]
Single layer GR	C_2_H_5_OH	1 mg/mL	48		[287]
GRO	C_2_H_5_OH + SAE20W-50 oil	-	-	60 to 80	[298]
GR	Engine oil	0.025 wt%	∼80	∼33	[291]
Polyacrylamide-grafted-f-GRO	Water	0.2 to 1.0 wt%	46 to 55	13 to 37	[313]
Octadecylamine fGR	C_16_H_34_	0.06 wt%	26	9	[317]
r-GRO nanosheets	PEG 200	0.03 mg/mL	38	55	[304]
GR nanoplatelets	Palm-oil TMP ester + PAO	0.05 wt%	5	15	[296]
Urea-modified fluorinated GR	Water	1 mg/mL		64.4	[322]
Cu nanoparticles/PDA f-GRO nanosheets	Soyabean oil	0.1 wt%	57	27	[319]

## Data Availability

Not applicable.

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
