# Peer review of "Graphene: A Path-Breaking Discovery for Energy Storage and Sustainability"

_materials, 2022, doi:10.3390/ma15186241_

Round 1

Reviewer 1 Report

In this article "Graphene: A path-breaking discovery for energy storage and 2 sustainability", authors mentioned about the graphene derivatives and their applications in various fields. The manuscript consists of a lot of data However, the following points should be address before publication.

The abstract looks like the introduction Hence should be result oriented.

In the caption of figure, avoid using the short form of any term.

In introduction section, few recent references would be helpful to support the content like J. Compos. Sci. 2021, 5, 181. https://doi.org/10.3390/jcs5070181, Polymers 14 (4), 845, Carbon 2017, 124, 57–63, Journal of Cleaner Production 318, 128603.

Actually the article is looking like an essay type (very lengthy) not a scientific article, few things can be helpful as follows:

In synthesis part, add one figure of demonstration for better understanding.

In each part/section, add few sub-headings to increase the interest of readers.

Add few lines in each subheading by your own to interpret the previous studies.

In each part/section, add few related figures.

The conclusion must precise and attractive.

Please check the grammatical error throughout the manuscript to uplift the standard of the article.

Please check the spacing error throughout the manuscript.

Author Response

Response to the Reviewer 1

Comment 1

The abstract looks like the introduction hence should be result oriented

Author response:

The authors are highly obliged to the honorable reviewer for the in-depth analysis of the manuscript and are thankful for the valuable comment.

As recommended by the respected reviewer, the abstract section has been modified in terms of background, importance and applications of graphene in diverse disciplines.

Comment 2

In the caption of figure, avoid using the short form of any term.

Author response:

Thanks for the comments. In the revised manuscript, the authors have removed the abbreviation in captions.

 Comment 3

In introduction section, few recent references would be helpful to support the content like J. Compos. Sci. 2021, 5, 181. https://doi.org/10.3390/jcs5070181, Polymers 14 (4), 845, Carbon 2017, 124, 57–63, Journal of Cleaner Production 318, 128603.

Author response:

The authors are grateful to the distinguished reviewer for valuable comment. The related relevant reference have been added in the revised manuscript (Ref. 5, 215).

Comment 4

In synthesis part, add one figure of demonstration for better understanding.

Author response:

Many thanks for the insightful comment. A total of 11 figures have been added in the revised manuscript demonstrating the application of graphene and graphene related derivatives in different fields.

Comment 5

In each part/section, add few sub-headings to increase the interest of readers.

Author response:

The authors are grateful to the distinguished reviewer for his thorough examination of the manuscript and for his constructive comments. As per the suggestion, sub-headings have added in almost all the sections of the manuscript to increase the readability and interest of readers.

Comment 6

Add few lines in each subheading by your own to interpret the previous studies.

Author response:

Many thanks for the comment. As per the suggestion, few introductory lines in each sub-headings have been added in the revised manuscript to interpret the results.

Comment 7

In each part/section, add few related figures.

Author response:

Many thanks for the insightful comment.

A total of 11 figures have been added in the revised manuscript demonstrating the application of graphene and graphene related derivatives in different fields.

Comment 8

The conclusion must precise and attractive.

Author response:

Comment 9

Please check the grammatical error throughout the manuscript to uplift the standard of the article.

Author response:

Thanks for the comments, all the grammatical mistakes and typo errors have been removed in the revised manuscript.

Comment 10

Please check the spacing error throughout the manuscript.

Author response:

Thanks for the comments, all the spacing errors have been fixed in the revised manuscript.

Reviewer 2 Report

The paper is a review article on the study of Graphene with focus on applications for energy and sustainability.  In the paper applications of Graphene are described in detail.  Review articles on applications are rare, so the paper could be published.  I have a comment on the manuscript.  There are two many abbreviations.  For example, PANI (polyaniline), PPY (polypyrrole), MC (mesoporous),  ... , are standard ones?  GRO may be graphene oxide.  I recommend that the number of abbreviations is reduced

Author Response

Response to the Reviewer 2

Comment 1

The paper is a review article on the study of Graphene with focus on applications for energy and sustainability.  In the paper applications of Graphene are described in detail.  Review articles on applications are rare, so the paper could be published. 

Author response:

The authors are highly obliged to the honorable reviewer for recommending our work to be published in the esteemed journal.

Comment 2

I have a comment on the manuscript.  There are too many abbreviations.  For example, PANI (polyaniline), PPY (polypyrrole), MC (mesoporous),  ... , are standard ones?  GRO may be graphene oxide.  I recommend that the number of abbreviations is reduced.

Author response:

Many thanks to the valuable reviewer for constructive comment. The authors have reduced the abbreviations and use the only ones which are required to increase the readability and interest of readers.

Reviewer 3 Report

This manuscript is a review paper about derivatives and morphologies of graphene in various energy-related applications. However, I found this review paper doesn’t contain any data/figures from the reviewed article, making it very unreadable. The authors should consider acquiring the copyright from the publishers of the reviewed papers and rewrite the entire manuscript with the data/figures organized. Besides, Fig. 1 is a reproduction of Ref. 2. In this case, the authors should get the copyright for reproduction from the publisher and use their original figure. For the current version, I don’t recommend to be published in Materials.

Author Response

Comment 1

This manuscript is a review paper about derivatives and morphologies of graphene in various energy-related applications. However, I found this review paper doesn’t contain any data/figures from the reviewed article, making it very unreadable. The authors should consider acquiring the copyright from the publishers of the reviewed papers and rewrite the entire manuscript with the data/figures organized. Besides, Fig. 1 is a reproduction of Ref. 2. In this case, the authors should get the copyright for reproduction from the publisher and use their original figure. For the current version, I don’t recommend to be published in Materials.

Author response:

The authors respect the observation of honorable reviewer for the in-depth analysis of the manuscript and are thankful for the valuable comment.

The revised manuscript has been thoroughly modified to enhanced the quality of previously submitted manuscript. All suggestions of other reviewers has also be incorporated. A total of 11 figures have been added in the revised manuscript demonstrating the application of graphene and graphene related derivatives in different fields. Also, the authors have reproduced the figures by getting the permission from the publisher.

Reviewer 4 Report

The review paper by Goyal et. al. gives an overview of different graphene-based materials in energy sustainability and environment friendly applications. This paper presents the in-depth review on the exploration of deploying diverse derivatives and morphologies of graphene in various energy related and environment benign applications, such as in solar cells, fuel cells, Li- ion and Na-ion rechargeable batteries and supercapacitor applications.

Overall, the review article is well written, organized and highlights all the key points related to employing high quality graphene-based derivatives for green and environmentally friendly applications. I do agree with the authors that more experimental investigations of graphene composites electrodes and membranes are needed to check the long-term effect on the performance of batteries and fuel cells to utilize it effectively as a potential energy saving material. I enjoyed reading the paper and believe that the review article might attract significant interest of the readership if the authors could provide a schematic view or figures of the examples reviewed through literature survey. My comments are given below

a)     It would be better if the authors could provide schematic representation (from the original references) depicting the synthesis methods discussed in the article.

b)     Similarly for general reading, the authors should also include figures (at least one example for each section) for use of graphene-based derivatives in supercapacitor applications, batteries and fuel cells. For e.g., in page 11 (line 429-430), the authors state that “Guo et al. [109] introduced a flexible integrated electrode based on layer-by-layer packed GR sheets entrapped SnO2-Co3O4 nanocubes. The obtained electrode showed excellent reversible capacity of 1665 mA h/g after 100 cycles at 100 mA/g. The readers might get a better idea if they could visualize the figures while reading the text.

c)      In the context of economical and environment-friendly alternatives for supercapacitor electrodes towards enhanced performances in metal-ion battery application, the authors should add a small comparison section in brief for graphene-based derivatives with nature-derived biodegradable organic supercapacitor electrodes (ChemSusChem 2020, 13, 2186 – 2204) in the manuscript.

Author Response

Comment 1

I enjoyed reading the paper and believe that the review article might attract significant interest of the readership if the authors could provide a schematic view or figures of the examples reviewed through literature survey.

Author response:

The authors are highly obliged to the honorable reviewer for recommending our work to be published in the esteemed journal.  As per the comment of the esteemed reviewer, a total of 11 figures have been added in the revised manuscript demonstrating the application of graphene and graphene related derivatives in different fields

Comment 2

 It would be better if the authors could provide schematic representation (from the original references) depicting the synthesis methods discussed in the article. Similarly for general reading, the authors should also include figures (at least one example for each section) for use of graphene-based derivatives in supercapacitor applications, batteries and fuel cells. For e.g., in page 11 (line 429-430), the authors state that “Guo et al. [109] introduced a flexible integrated electrode based on layer-by-layer packed GR sheets entrapped SnO2-Co3O4 nanocubes. The obtained electrode showed excellent reversible capacity of 1665 mA h/g after 100 cycles at 100 mA/g. The readers might get a better idea if they could visualize the figures while reading the text.

Author response:

Many thanks for this constructive comment. The authors have added the relevant schematic representation of graphene and graphene-based derivatives in diverse disciplines. Moreover, subheadings have been framed to increase the readability and interest of readers.

Comment 3

 In the context of economical and environment-friendly alternatives for supercapacitor electrodes towards enhanced performances in metal-ion battery application, the authors should add a small comparison section in brief for graphene-based derivatives with nature-derived biodegradable organic supercapacitor electrodes (ChemSusChem 2020, 13, 2186 – 2204) in the manuscript.

Author response:

The authors are highly obliged the esteemed reviewer for his insightful comment. The present paper is the review of graphene in various energy saving and environment friendly applications, comparison for graphane-based derivatives with nature-derived biodegradable organic supercapacitor electrodes will be explored in future work. Moreover, the mentioned relevant related work has been cited in the revised manuscript as Ref. 98.

Round 2

Reviewer 1 Report

can be accepted

Author Response

Many thanks to reviewer for accepting the manuscript. As suggested English language has been proof read and corrected where ever, it was required.

Reviewer 3 Report

The revised manuscript is well-organized, and I think the current manuscript is clearly written and can be published in Materials.

Author Response

many thanks to reviewer for accepting the manuscript. the revised manuscript is modified with corrected spellings and English language, wherever it was need.